# A Privacy-Preserving and Unified Federated Learning Framework for Trajectory Data Preparation

## Abstract

Trajectory data, which captures the movement patterns of people and vehicles over time and space, is crucial for applications such as traffic optimization and urban planning. However, issues such as noise and incompleteness often compromise data quality, leading to inaccurate trajectory analyses and limiting the potential of these applications. While Trajectory Data Preparation (TDP) can enhance data quality, existing methods suffer from two key limitations: (i) they do not address data privacy concerns, particularly in federated settings where trajectory data sharing is prohibited, and (ii) they typically design task-specific models that lack generalizability across diverse TDP scenarios. To overcome these challenges, we propose FedTDP, a privacy-preserving and unified framework that leverages the multi-task learning capabilities of Large Language Models (LLMs) for TDP in federated environments. Specifically, we: (i) design a trajectory privacy autoencoder for secure data transmission to protect data privacy with theoretical analysis, (ii) introduce a trajectory knowledge enhancer to develop TDP-oriented LLMs by improving model learning of TDP knowledge, and (iii) propose federated parallel optimization to enhance training efficiency by reducing data transmission and enabling parallel model training. Experiments on 6 real datasets and 10 mainstream TDP tasks demonstrate that FedTDP consistently outperforms 13 state-of-the-art baselines. All code and data are available at `https://anonymous.4open.science/r/FedTDP`.

## 1 Introduction

Trajectory data is typically represented as sequences of spatio-temporal points that describe the movement of objects, such as people [6] and vehicles [68]. Proliferation of GPS and location-based services has generated vast amounts of trajectory data [56, 74, 85], enabling various analytical applications, including route planning [89], crowd clustering [39], and traffic prediction [41]. However, trajectory data often suffers from significant quality issues due to sensor malfunctions, limited equipment precision, and transmission interruptions, leading to inconsistent [43], noisy [17], and missing values [9]. For instance, GPS location estimates in Uber [65] can be inaccurate by over 50 meters in densely populated, highly built-up urban areas. Such low-quality data undermines the reliability of trajectory analyses, limiting their practical applications. To address these issues, **Trajectory Data Preparation (TDP)**—which includes preprocessing and data mining such as data imputation [47], map matching [43], trajectory-user linking [7], anomaly detection [22], and trajectory recovery [44]—has become essential for improving data quality before analysis and application.

However, existing TDP methods face two key limitations that affect their privacy and generalizability. (i) *Previous studies have not addressed data privacy constraints.* According to government reports [20] and related studies [78, 45, 59], trajectory data is often collected or stored across multiple stations or organizations. Consequently, a moving trajectory may span several geographic regions, with each region's data collected by its respective signal station. For example, Fig. 1 illustrates the

trajectory data from GeoLife [86], a real-world dataset collected in Beijing, which shows six distinct colored regions, each storing its trajectory data separately. Due to legal constraints [23, 4, 10], the exchange of trajectory data across regions is prohibited. However, existing studies typically assume centralized data, which increases the risk of privacy breaches. (ii) *All previous studies are single-task approaches.* Specifically, these models are tailored to a single TDP task, such as data imputation or anomaly detection. When addressing multiple TDP tasks, a new model must be trained for each task, resulting in high computational cost, extended training time, and limited generalizability.

Motivated by these limitations, we pro-
pose a privacy-aware and generalized
framework for trajectory data prepara-
tion. (i) To protect data privacy, we in-
troduce Federated Learning (FL)[40, 31],
a privacy-preserving distributed learning
paradigm. FL has been widely applied in
domains such as urban computing [66]
and transportation management [73] to
address privacy concerns. For example,
MobiSpaces [46], a government-funded
project by the European Union, collabo-
rates with various transportation services
to support Mobility-as-a-Service using FL.
It provides a data governance platform for
processing raw trajectory data from public

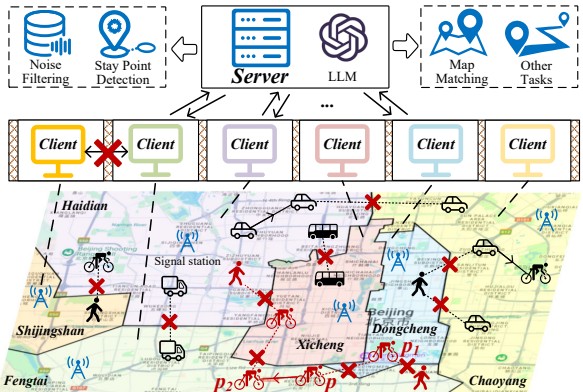

Figure 1: Federated trajectory data preparation

transportation, traffic sensors, and maritime vessels for decentralized analysis. As shown in Fig. 1, FL enables multiple **Clients** (i.e., regions) to collaboratively train a model on a **Server** while keeping trajectory data decentralized, thereby preserving the data privacy of each client, a new problem referred to as **Federated Trajectory Data Preparation** (F-TDP). (ii) Inspired by the powerful capabilities of Large Language Models (LLMs) [75, 42, 69], particularly their success in multi-task learning, we aim to develop a **TDP-oriented LLM** to support various TDP tasks. Overall, our goal is to leverage LLMs to create a **privacy-preserving and unified framework FedTDP** for trajectory data preparation in the federated learning environment. However, developing the FedTDP framework presents three key technical challenges that must be addressed.

*Challenge 1: How to safeguard trajectory data privacy in the FedTDP framework?* TDP tasks often necessitate considering the data context [22, 47, 43], which involves the exchange and sharing of data and demands collaborative processing across clients (i.e., cross-client TDP), raising privacy concerns. As shown in Fig. 1, if the data $p$ is missing, the Fengtai region needs to utilize the context of the missing data (i.e., $p_1$ and $p_2$) for data imputation [43, 71]. However, due to data privacy constraints, the Fengtai region cannot access $p_1$ from the Dongcheng region. Consequently, ensuring the privacy of trajectory data thus constitutes the first challenge the FedTDP framework must address.

*Challenge 2: How to develop a TDP-oriented LLM in FedTDP?* Existing LLMs perform poorly on TDP tasks due to several factors. First, they are primarily designed for text data [16, 34]. However, trajectory data exhibits unique spatio-temporal features [47, 18], such as temporal regularity and spatial dependency, which differ significantly from text data and are not inherently understood by LLMs. Besides, their pre-training relies largely on publicly available unsupervised corpora [15, 24], which capture only general textual knowledge. In contrast, TDP tasks involve intricate spatio-temporal relationships and patterns [43, 29] that are not included in these corpora. As a result, effectively training a TDP-oriented LLM represents the second challenge that FedTDP must overcome.

*Challenge 3: How to improve the training efficiency of the FedTDP framework?* Due to the limited computational resources and storage capacities of clients, directly deploying and training LLMs locally on clients is infeasible [62, 79, 26]. As a result, LLMs are typically hosted on servers, requiring clients (i.e., regions) to transmit their local data to the server for TDP processing. This introduces storage burdens on the server and wastes computational resources on the client. Additionally, LLMs often contain a large number of parameters, and even with techniques like Parameter-Efficient Fine-Tuning (PEFT) [28], training a TDP-oriented LLM remains highly time- and resource-intensive. Therefore, enhancing training efficiency represents the third challenge that FedTDP must address.

**Contributions.** To address the challenges outlined above, we first introduce the **Small Language Model (SLM)**, a compact version of the server's LLM, which is deployed on each client for local TDP.

This approach leverages the computational resources of clients and reduces the server's workload, enabling distributed computation within the FedTDP framework. To address ***Challenge 1***, we propose a **Trajectory Privacy AutoEncoder (TPA)**, which encodes trajectory data into spatio-temporal embeddings for transmission, rather than sending the raw data. This ensures data privacy while preserving the spatio-temporal correlations essential for TDP tasks. Besides, we develop a decentralized secret-sharing method to safeguard against trajectory data recovery or inference through embedding [61, 8, 30] and gradient [67, 84, 87] inversion attacks with the theoretical privacy analysis in Appendix C.5. To address ***Challenge 2***, we design a **Trajectory Knowledge Enhancer (TKE)**, which helps both the SLM and LLM understand trajectory data and learn the specific knowledge required for TDP tasks. This enhances the model's ability to learn TDP-related patterns while reducing the number of parameters. To tackle ***Challenge 3***, we introduce **Federated Parallel Optimization (FPO)** to improve training efficiency. Specifically, FPO decomposes the federated training between the server and clients through split learning, employs alternating optimization to minimize data transmission, and accelerates training via parallel execution. Finally, experiments on 6 real-world datasets demonstrate that the proposed FedTDP framework outperforms 13 state-of-the-art baselines, achieving a performance improvement from **4.84% to 45.22%** across 10 mainstream TDP tasks.

## 2 Preliminary

The frequently used notations and descriptions in this paper are shown in **Appendix B**.

**Definition 1 (Spatio-Temporal Point)**. *A spatio-temporal point is represented as $p = \langle l, t \rangle$, where $l = (lon, lat)$ is a tuple of longitude and latitude location coordinates, and $t$ refers to the observed time associated with this spatio-temporal point.*

**Definition 2 (Trajectory)**. *A trajectory comprises chronological spatio-temporal points, denoted as $T = \{p_1, p_2, \ldots\}$, which is typically represents the movement of a user. In addition, a trajectory can be segmented into multiple sub-trajectories, denoted as $T = \{ST^{(1)}, ST^{(2)}, \ldots\}$.*

**Definition 3 (Data Silo)**. *A data silo $S$ has its own collected trajectory dataset $D$. In federated learning, a data silo $S$ is represented as a client $C$, typically a regional data storage platform or institution, responsible for the collection and management of trajectory data within that region. Specifically, a trajectory $T = \{p_1, p_2, \ldots\}$ is segmented into sub-trajectories based on the geographic locations, denoted as $T = \{ST^{(C_1)}, ST^{(C_2)}, \ldots\}$, where sub-trajectory $ST^{(C_i)}$ is stored in client $C_i$.*

**Problem Formulation (F-TDP)**. Given the server's LLM $\theta_{LLM}$ and the trajectory dataset $\mathcal{D} = \{D_1, D_2, \ldots\} \rightarrow \{T_1, T_2, \ldots\}$ of all clients $\mathcal{C} = \{C_1, C_2, \ldots\}$, where client $C_i$ holds dataset $D_i$, F-TDP is to employ $\theta_{LLM}$ on $\mathcal{D}$ for performing various trajectory data preparation tasks, where collected trajectories $D_i$ of client $C_i$ cannot be shared and exchanged to the server and other clients:

$$F\text{-}TDP(\mathcal{D}) = \theta_{LLM}(T_i), T_i = \{ST_i^{(C_1)}, ST_i^{(C_2)}, \ldots\}, \tag{1}$$

where $\theta_{LLM}(T_i)$ is the result of $\theta_{LLM}$ on the trajectory $T_i$, with different forms of output depending on the TDP task, such as the cleaned trajectory, points, or classification result.

## 3 Trajectory Data Preparation Task

We demonstrate all major types of TDP tasks, with the rough processing shown in **Appendix B**.

**T-1: Anomaly Detection (*AD*).** It aims to detect trajectories that deviate significantly from typical movement behaviors. These anomalies could result from unusual user behavior, errors in data collection, or potential malicious activities.

**T-2: Trajectory Imputation (*TI*).** It aims to reconstruct a complete trajectory by estimating the missing points based on available spatio-temporal points. This often occurs when GPS signals are lost or data collection is interrupted.

**T-3: Noise Filtering (*NF*).** It aims to identify and remove irrelevant spatio-temporal points that deviate from a trajectory. These noisy points can result from GPS inaccuracies, signal interference, or sensor malfunctions.

**T-4: Stay Point Detection (*SPD*).** It aims to identify locations where a moving object remains within an area for a certain period of time. A stay point typically represents a place of interest, such as a rest stop, home, or office.

**T-5: Map Matching (*MM*).** It aims to map the spatio-temporal point to the most probable segment in the road network. This is often the case when there is a deviation in the collected GPS position.

146 **T-6: Trajectory-User Link (*TUL*).** It aims to link an anonymous trajectory with its corresponding
147 user. These trajectories are often collected without any user-identifying information.

148 **T-7: Travel Mode Identification (*TMI*).** It aims to identify the travel mode based on the moving
149 pattern of trajectory, which is walking, biking, taking the bus, or driving a car.

150 **T-8: Trajectory Simplification (*TSim*).** It aims to reduce the number of spatio-temporal points in a
151 trajectory while preserving its essential shape and features.

152 **T-9: Trajectory Segmentation (*TSeg*).** It aims to divide a trajectory into meaningful segments based
153 on specific criteria such as stay points or travel modes.

154 **T-10: Trajectory Recovery (*TR*).** It aims to reconstruct a complete trajectory from partially observed
155 or incomplete spatio-temporal points. This often occurs when some parts of the trajectory are missing
156 or unobserved.

# 4 Our Approach

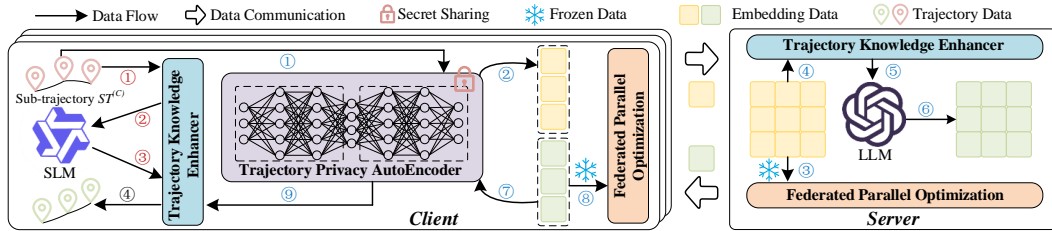

Figure 2: The overview of our framework

158 Fig. 2 shows an overview of the FedTDP framework, which involves a server and multiple clients.
159 FedTDP consists of three modules, i.e., Trajectory Privacy AutoEncoder (TPA), Trajectory Knowledge
160 Enhancer (TKE), and Federated Parallel Optimization (FPO). To enable distributed computing of
161 FedTDP, we first introduce the Small Language Model (SLM), a small-scale version of the server's
162 Large Language Model (LLM), which is deployed on each client for local TDP, to leverage clients'
163 computational resources and reduce the server's workload. Specifically, suppose the data context of
164 TDP on the client's sub-trajectory $ST^{(C)}$ does not involve data from other clients for joint processing.
165 In that case, the locally deployed SLM is used for local TDP, or $ST^{(C)}$ must be uploaded to the server
166 and use the LLM for cross-client TDP. The overall process is as follows. For the local TDP, TKE
167 generates the TDP prompt as input for SLM (①–②). Next, TKE enhances the TDP knowledge to get
168 the final result (③–④). For the cross-client TDP, TPA first encodes trajectory data and transmits the
169 encoded embeddings to the server (①–②). Then, FPO freezes the data transmitted from clients (③)
170 and TKE generates the TDP prompt as input for LLM (④–⑤). Next, TPA decodes results outputted
171 by the server's LLM (⑥–⑦). Finally, FPO freezes the data transmitted from the server (⑧) and TKE
172 enhances the TDP knowledge to get final results (⑨–④).

## 4.1 Trajectory Privacy AutoEncoder

174 **Design Motivation**. As aforementioned, F-TDP involves the joint processing of data from multiple
175 clients, i.e., cross-client TDP, necessitating data exchange and sharing. Consequently, safeguarding
176 the privacy of trajectory data becomes essential. Although differential privacy [14] can be applied
177 to ensure data privacy, it requires adding noise to the data, which diminishes its utility and reduces
178 model accuracy. In contrast, FedTDP proposes a Trajectory Privacy AutoEncoder (TPA) to protect
179 trajectory data privacy while maintaining spatio-temporal correlations.

180 Specifically, the TPA module employs an encoder-decoder architecture that encodes trajectory
181 data $T = \{p_1, p_2, \ldots\}$ into embeddings $E = \{e_1, e_2, \ldots\}$, where each spatio-temporal point $p_i$ is
182 independently encoded as $e_i = \theta_{Enc}(p_i)$. Then, these clients' embeddings are transmitted to the
183 server for aggregation $\mathcal{E} = \bigcup_{i=1}^{|\mathcal{C}|} E_i$, preserving both intra-client and inter-client spatio-temporal
184 dependencies, which helps the LLM to capture spatio-temporal relationships in the trajectory data.
185 Next, the server splits and distributes results $\tilde{\mathcal{E}} = \{\tilde{e_1}, \tilde{e_2}, \ldots\}$ outputted by the LLM to clients,
186 where the decoder reconstructs the estimated trajectory $\tilde{T} = \{\tilde{p_1}, \tilde{p_2}, \ldots\}$ through $\tilde{p_i} = Dec(\tilde{e_i})$.
187 Here, TPA is implemented as a lightweight three-layer MLP (Multi-Layer Perception) [55] with
188 GELU [25] activation, 32 embedding dimensions, and 256 hidden dimensions, which does not
189 introduce significant computational overhead, as also proved in the ablation study (see Section 5.2).

However, merely using embeddings for transmission cannot safeguard data privacy completely in FL, as attackers can recover the raw data by embedding [61, 8, 30] and gradient [88, 84, 87] inversion attacks during TPA model aggregation. Specifically, traditional FL model aggregation, which exchanges client gradients and aggregated parameters, are vulnerable to these attacks. While homomorphic encryption [54] and differential privacy [14] offer solutions, they introduce computational overhead or degrade model accuracy. In contrast, we propose a decentralized aggregation approach based on secret sharing [58], achieving secure TPA aggregation without compromising efficiency or accuracy. Initially, each client pair $(C_i, C_j)$ generates a shared secret key $sk_{i,j} = sk_{j,i}$ stored locally, respectively. Then, the TPA parameters are partitioned into $|\mathcal{C}|$ parameter blocks $\{P^{(1)}, P^{(2)}, \ldots\}$. For aggregation, the client $C_i$ masks its parameter block using secret keys $\{sk_{i,0}, sk_{i,1}, \ldots\}$ determined with the other clients to mask parameter blocks, adding $sk_{i,j}$ if $i > j$ or subtracting it if $i < j$, as shown below:

$$\tilde{P}_i^{(k)} = P_i^{(k)} + \sum_{j=1 \& j \neq i}^{|\mathcal{C}|} a_{i,j} * sk_{i,j} \, , \;\; a_{i,j} = \left\{ \begin{array}{l} 1, \;\; i < j \\ -1, \;\; i > j \end{array} \right. , \tag{2}$$

where client $C_i$ holds the parameter block $P_i^{(k)}$ and $\tilde{P}_i^{(k)}$ is the mask parameter block.

**Theorem 1.** *Given the mask parameter blocks $\{\tilde{P}_1^{(k)}, \tilde{P}_2^{(k)}, \ldots\}$ from all clients, the result of aggregating them is equal to the result of aggregating raw parameter blocks $\{P_1^{(k)}, P_2^{(k)}, \ldots\}$ for all clients directly, as formally shown below:*

$$\sum_{i=1}^{|\mathcal{C}|} \tilde{P}_i^{(k)} = \sum_{i=1}^{|\mathcal{C}|} P_i^{(k)} \tag{3}$$

*Proof.* The detailed proofs of Theorem 1 are provided in **Appendix C.1**. $\qquad\square$

According to Theorem 1, the client $C_k$ can obtain the aggregation result $\overline{P}^{(k)}$ of the parameter block $P^{(k)}$ by aggregating the mask parameter blocks transmitted from clients, as formally shown below:

$$\overline{P}^{(k)} = \frac{1}{|\mathcal{C}|} \sum_{i=1}^{|\mathcal{C}|} \tilde{P}_i^{(k)} = \frac{1}{|\mathcal{C}|} \sum_{i=1}^{|\mathcal{C}|} P_i^{(k)} \tag{4}$$

Finally, the aggregated parameter block is broadcast to clients for the TPA model updates.

## 4.2  Trajectory Knowledge Enhancer

**Design Motivation**. Since existing LLMs are designed for text data and contain only general textual knowledge [16, 34, 15], they cannot be directly applied to trajectory data and TDP tasks. Although a few spatio-temporal LLMs [38, 82, 36] have been proposed, none of them have considered TDP. In contrast, to develop a TDP-oriented LLM, FedTDP designs Trajectory Knowledge Enhancer (TKE) that consists of trajectory prompt engineering, trajectory offsite-tuning, LoRA sparse-tuning, and bidirectional knowledge learning, to enhance the model learning abilities on TDP knowledge.

**i) Trajectory Prompt Engineering** To help the SLM and LLM understand trajectory data and learn TDP knowledge, TKE designs a trajectory instruction paradigm to generate the TDP prompt, defined as (***Task***, ***Data***, ***Information***, ***Format***). Specifically, ***Task*** is the textual instruction consisting of the task name and the task description, as listed in Section 3. ***Data*** is the input trajectory data, either as trajectory data $T = \{p_1, p_2, \ldots\}$ to the SLM for local TDP or embeddings $E = \{e_1, e_2, \ldots\}$ to the LLM for cross-client TDP. ***Information*** is the optional trajectory context (e.g., road network, weather) from public sources such as OpenStreetMap [49] and weather services [50], to enhance the model's ability to perform TDP tasks. ***Format*** is the task-specific output format, such as classification results for TDP tasks including AD, TUL, and TMI, trajectories for TDP tasks including TI, NF, TSim, TSeg, MM, and TR, and spatio-temporal points for the SPD task. A few examples of TDP tasks using the trajectory prompt engineering are shown in **Appendix C.2**.

**ii) Trajectory Offsite-Tuning.** To enhance the learning capabilities of the SLM in clients, TKE employs the LLM to assist it in learning trajectory knowledge by trajectory off-site tuning. Specifically, inspired by the offsite-tuning [70], we divide the LLM into two components, denoted as $\theta_{LLM} = [\mathcal{A}, \mathcal{F}]$. Here, the adapter $\mathcal{A}$ is the last few layers of the LLM to specialize general features for specific tasks, enabling task-specific feature mapping and decision making. Besides, the foundation $\mathcal{F}$ is the remaining layers excluding $\mathcal{A}$, to extract general data features, transforming raw inputs into meaningful representations. Initially, it dispatches the server's adapter $\mathcal{A}$ to the client as the final few layers to be integrated into the client's SLM. Consequently, the SLM is composed of two components,

denoted as $\theta_{SLM} = [\mathcal{A}, \mathcal{F}']$, where $\mathcal{F}'$ is the foundation of the SLM. Subsequently, the SLM employs LoRA to reduce the number of parameters in the adapter and then transmits the fine-tuning adapter to the server for aggregation and updates. Note that, rather than directly transferring the trained LLM's adapter to SLM, it utilizes and trains it to augment the SLM's learning capacity during training.

**iii) LoRA Sparse-Tuning.** To reduce the number of training parameters, TKE proposes LoRA sparse-tuning, as shown in Fig. 3. According to works on sparsity [1, 13, 80], more significantly varying parameters have a greater contribution to model convergence. Therefore, we only choose the layer in the SLM where the LoRA parameter change rate is the top $m$ for

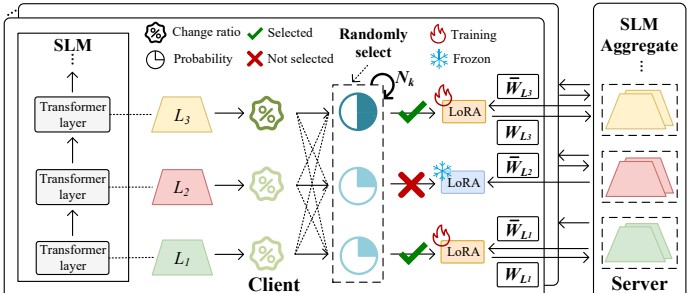

Figure 3: LoRA sparse-tuning

training. Specifically, the client calculates the ratio of the LoRA parameters change rate of each layer to the global LoRA parameters change rate of all $N$ layers ("ratio" for short), as shown below:

$$R^{(r)}(L_i) = \frac{CR^{(r)}(L_i)}{\sum_{j=1}^{N} CR^{(r)}(L_j)} , \tag{5}$$

where $CR^{(r)}(L_i)$ is the LoRA parameters change rate of layer $L_i$ at round $r$, as shown below:

$$CR^{(r)}(L_i) = |\frac{L_i^{(r)} - L_i^{(r-1)}}{L_i^{(r-1)}}| \tag{6}$$

Then, we randomly select $N_m = \lfloor m * N \rfloor$ layers to participate in the next round of the SLM training.

**Theorem 2.** *Given the ratio $R^{(r)}(L_i)$ of layer $L_i$ in training round $r$ and the number of layers $N_m$ to be trained, the probability $Pr^{(r+1)}(L_i, N_m)$ of layer $L_i$ to train in the next round $r + 1$ is shown:*

$$Pr^{(r+1)}(L_i, N_m) = R^{(r)}(L_i) + \sum_{j_1=1}^{N} \frac{R^{(r)}(L_i) * R^{(r)}(L_{j_1})}{1 - R^{(r)}(L_{j_1})} + \dots$$

$$+ \sum_{j_1=1}^{N} \cdots \sum_{j_{N_m}=1}^{N} \frac{R^{(r)}(L_i) * R^{(r)}(L_{j_1}) * \dots * R^{(r)}(L_{j_{N_m}})}{1 - R^{(r)}(L_{j_1}) - \dots - R^{(r)}(L_{j_{N_m}})} , \tag{7}$$

$$j_1 \neq \dots \neq j_{N_m} \neq i ,$$

*Proof.* The detailed proofs of Theorem 2 are provided in **Appendix C.3**. □

According to Theorem 2, it chooses the training layers based on their probability at each training round. Finally, the client uploads the LoRA parameters of the trained layers to the server for aggregation, and the server assigns different weights to the parameters based on the number of clients involved in training on these layers, as formally shown below:

$$\overline{W}_{L_i}^{(r)} = \frac{(|\mathcal{C}| - |\mathcal{C}'|) * \frac{\sum_{j=1}^{|\mathcal{C}'|} n_j * W_{L_i,j}^{(r)}}{\sum_{j=1}^{|\mathcal{C}'|} n_j} + \overline{W}_{L_i}^{(r-1)}}{|\mathcal{C}| - |\mathcal{C}'| + 1} , \tag{8}$$

where $W_{L_i,j}^{(r)}$ is the LoRA parameters of layer $L_i$ sent by client $C_j$ at training round $r$, $\overline{W}_{L_i}^{(r)}$ is the aggregated LoRA parameters, and $|\mathcal{C}'|$ is the number of clients that have trained layer $L_i$.

**iv) Bidirectional Knowledge Learning.** To improve the model learning capabilities, TKE develops bidirectional knowledge learning to enhance their TDP knowledge. Specifically, in order for the SLM to learn useful TDP knowledge in the complex output space of the LLM, it aligns the SLM's output with LLM's high frequency output using the inverse Kullback–Leibler (KL) divergence [33]:

$$\min_{\theta_{SLM}} D_{KL}(P_{\theta_{SLM}} || P_{\theta_{LLM}}) = \sum_{T} P_{\theta_{SLM}}(T) \log(\frac{P_{\theta_{SLM}}(T)}{P_{\theta_{LLM}}(T)}) \tag{9}$$

where $P_{\theta_{SLM}}$ and $P_{\theta_{LLM}}$ are the output distribution of the SLM and LLM, respectively. Besides, since the SLM can access raw trajectory data, it aligns the LLM's output with the SLM's output using the forward KL divergence, which enables the LLM to learn the trajectory knowledge of the SLM:

$$\min_{\theta_{LLM}} D_{KL}(P_{\theta_{SLM}} || P_{\theta_{LLM}}) = \sum_{T} P_{\theta_{SLM}}(T) \log(\frac{P_{\theta_{SLM}}(T)}{P_{\theta_{LLM}}(T)}) \tag{10}$$

### 4.3 Federated Parallel Optimization

**Design Motivation.** Since the proposed framework employs a federated training process, a significant amount of data must be transferred between the client and server during each training round, resulting in substantial communication overhead that reduces training efficiency. Additionally, gradient backpropagation [57] is required between the client and server in every train-

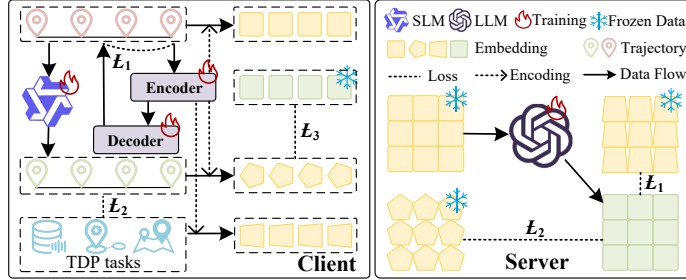

Figure 4: Federated parallel optimization

ing round. To improve training efficiency, FedTDP introduces Federated Parallel Optimization (FPO), which utilizes split learning, alternating optimization, and parallel training to reduce data transmission and enhance the training parallelism. The overall process of the FPO module is shown in Fig. 4.

First, to enable the simultaneous training of the client and server, it employs split learning [21] to decompose the federated training process into client and server training. Specifically, the client is responsible for the training of the TPA model (i.e., the encoder and decoder) and SLM, while the server manages the training of the LLM. Besides, to reduce data transmission, it utilizes alternating optimization [45] to freeze the data required by the client and server, respectively. During training, the server freezes the embeddings uploaded by the client for the LLM training, while the client freezes the results outputted by the server's LLM for the TPA model and SLM training. Finally, to enhance the training parallelism, it uses parallel training to optimize several objectives in parallel. Specifically, the client focuses on three optimization objectives: (i) minimizing the reconstruction loss $\mathcal{L}_1$ of the TPA model, (ii) reducing the inverse KL loss $\mathcal{L}_2$ between SLM and LLM outputs, and (iii) minimizing the loss $\mathcal{L}_3$ between SLM outputs and labels. On the other hand, the server has two optimization objectives: (i) minimizing the forward KL loss $\mathcal{L}_1$ between LLM and SLM outputs, and (ii) reducing the loss $\mathcal{L}_2$ between LLM outputs and labels.

The training process and privacy analysis of FedTDP are shown in **Appendices C.4** and **C.5**.

## 5 Experiment

Table 1: The evaluated trajectory data preparation tasks

| Type | Category | Task | Dataset |
|------|----------|------|---------|
| **Seen** (seen in training) | Data Cleaning | Anomaly Detection (**AD**) Trajectory Imputation (**TI**) | Geolife |
| | Data Matching | Map Matching (**MM**) Trajectory User Linking (**TUL**) | |
| | Data Annotation | Travel Mode Identification (**TMI**) | |
| | Data Reduction | Trajectory Simplification (**TSim**) | |
| | Data Augmentation | Trajectory Recovery (**TR**) | |
| **Unseen** (unseen in training) | Data Cleaning | Anomaly Detection Trajectory Imputation Noise Filtering (**NF**) Stay Point Detection (**SPD**) | Porto T-Drive |
| | Data Matching | Map Matching Trajectory User Linking | Tencent Gowalla |
| | Data Annotation | Travel Mode Identification | SHL |
| | Data Reduction | Trajectory Segmentation (**TSeg**) Trajectory Simplification | T-Drive |
| | Data Augmentation | Trajectory Recovery | |

**Tasks and Datasets.** We evaluate the framework by the 10 mainstream tasks and 6 datasets in Table 1, which are widely studied in TDP communities [43, 47, 27]. For the underlined seen task, we conduct experiments using the **GeoLife** [86] dataset, which was collected from April 2007 to August 2012. It contains various quality issues such as positional inaccuracies, data noise, and lower precision, which makes it suitable for various tasks. For the underlined unseen task, the following datasets are used. (i) **Porto** [51] was collected in Porto from July 2013 to June 2014 with 442 taxis, which contain quality issues such as anomalies and missing data. (ii) **T-Drive** [76] was collected in Beijing in February 2008 with 10,357 trajectories, which contain quality issues such as noisy and incomplete points.

(iii) **Tencent** [43] was collected in Beijing city for 3 months, which contains quality issues such as inaccurate points due to the low sampling rate. (iv) **Gowalla** [11] was collected in the social network from January to June 2010 with 6,442,890 check-in data from 10,336 users. (v) **SHL** [60] was collected by the University of Sussex over 7 months in 2017 from 3 users, which contains various travel and movement modes. More details of these datasets are provided in **Appendix D.1**.

**Baselines.** We compare FedTDP with (i) **none-LLM methods** including ATROM [22] (for anomaly detection task), Kamel [47] (for trajectory imputation task), GraphMM [43] (for map matching task), AttnTUL [7] (for trajectory-user link task), Estimator [27] (for travel mode identification task), S3 [18] (for trajectory simplification task), and LightTR [44] (for trajectory recovery task), which are the leading approaches in their respective research tasks; (ii) three SOTA **LLM-based table data preparation methods**, namely FM4DP [48], MELD [72], and TableGPT [35]; and (iii) three SOTA **LLM-based spatio-temporal data analysis methods**, including PromptGAT [12], UniST [77], and UrbanGPT [37]. More details of these baselines are provided in **Appendix D.2**.

**Implementations.** Synchronized Euclidean Distance (SED) is used for the trajectory simplification task, while $F_1$ scores are used for other tasks. The lower the SED and the higher the $F_1$ score, the better the performance. Besides, we use the running time and communication size to evaluate the efficiency. All baselines run under their optimal settings. Besides, FedTDP can protect data privacy with the TPA module, while other baselines do not consider data privacy in F-TDP. To solve F-TDP, one alternative approach for baselines is to employ differential privacy [14]. Specifically, clients apply differential privacy to perturb local trajectory data before transmitting it to the server. Therefore, to safeguard data privacy and ensure fairness in experiments, we extend baselines combined with this optional approach to solve the F-TDP problem. Moreover, all experiments are conducted in the federation with 9 nodes, one as a server and the other 8 nodes as clients, each equipped with two Intel Xeon CPU E5-2650 12-core processors, two GeForce RTX 3090, and 100 MB/s internet.

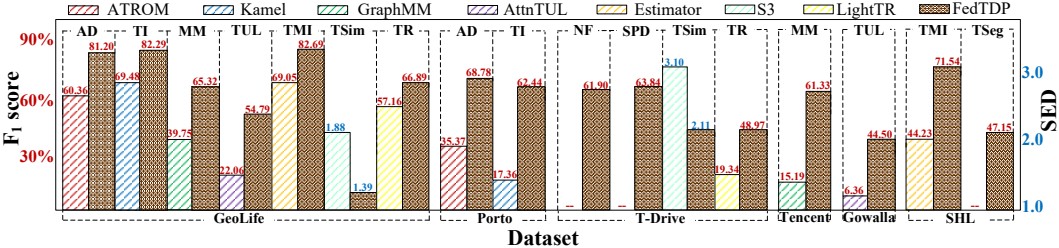

Figure 5: The performance of FedTDP and none-LLM trajectory data preparation methods

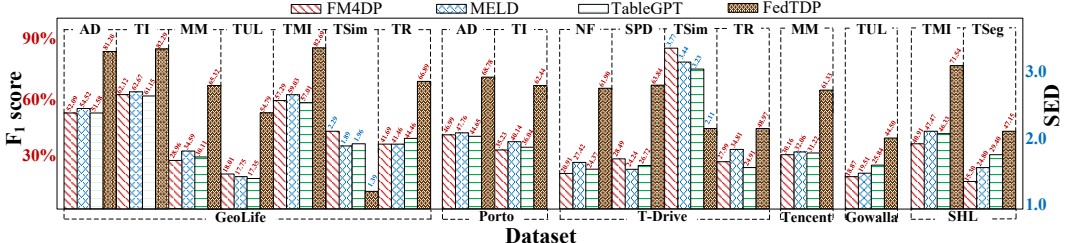

Figure 6: The performance of FedTDP and LLM-based table data preparation methods

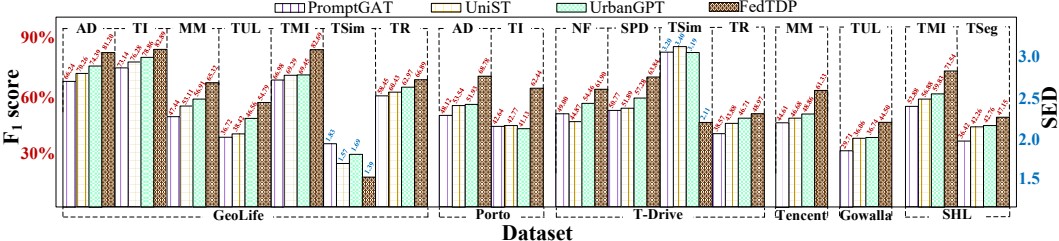

Figure 7: The performance of FedTDP and LLM-based spatio-temporal data analysis methods

## 5.1 Overall Performance

We present the overall performance comparison between the FedTDP framework with various SOTA baselines across different datasets and tasks. First, as shown in Fig. 5 (where the dash "– –" denotes

tasks that are not supported), FedTDP demonstrates the best performance and robust generalization with an improvement of at least **18.38%** compared with non-LLM TDP methods, highlighting its effectiveness in addressing the F-TDP problem. Besides, the superior performance achieved by FedTDP on unseen TDP tasks also demonstrates its strong generalization. Second, as shown in Fig. 6, compared with SOTA LLM-based table data preparation, FedTDP achieves the best performance across different datasets and tasks with an improvement of at least **32.26%**. Third, as shown in Fig. 7, compared with SOTA LLM-based spatio-temporal data analysis methods, FedTDP also shows the best performance with an improvement of **4.84% to 45.22%**. We attribute these improvements to the developed TDP-oriented LLM and SLM in the FedTDP framework.

## 5.2 Ablation Study

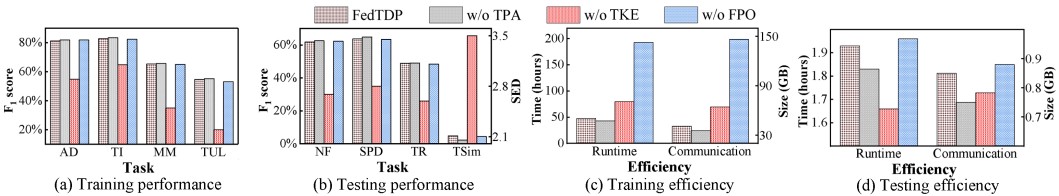

Figure 8: The ablation study

We evaluate the effectiveness of each module in the FedTDP framework by systematically removing one at a time, with the following configurations: FedTDP without Trajectory Privacy AutoEncoder (w/o TPA), without Trajectory Knowledge Enhancer (w/o TKE), and without Federated Parallel Optimization (w/o FPO). The results are shown in Fig. 8. First, the performance of FedTDP is slightly degraded compared to w/o TPA, as TPA can not fully capture the spatio-temporal information of the trajectory data, leading to a marginal performance decline when using TPA. Besides, FedTDP has a slight increase in runtime and communication costs because TPA transmits higher-dimensional embedding data instead of three-dimensional spatio-temporal points, introducing greater communication size and runtime when TPA is employed. However, to safeguard data privacy, the use of TPA in the FedTDP framework is essential. Second, the performance of FedTDP improves dramatically compared to w/o TKE, with at least **27.52%** improvement. This is because TKE enhances the model's learning abilities on TDP knowledge to develop the TDP-oriented LLM and SLM. Additionally, FedTDP has lower runtime and communication costs during training, since the TKE module can reduce the number of parameters that need to be trained and transmitted, which speeds up the model training. Finally, the performance of FedTDP does not change significantly compared to w/o FPT, but its training runtime and communication overhead are significantly reduced by almost **4 times** less. This reduction is because FPO can reduce data transmission and improve training efficiency.

## 5.3 More Experiments

We conduct more experiments to comprehensively evaluate FedTDP, in terms of model generalization, model base, efficiency, and hyperparameter sensitivity: i) **Appendix D.3** evaluates FedTDP's generalization in different numbers of training tasks, where the lower the number of seen tasks, the worse the model accuracy is of FedTDP in TDP tasks. ii) **Appendix D.4** evaluates the impact of various model bases on FedTDP, where Llama [64] achieves optimal performance in most tasks for the LLM, while GPT3-Small [5] demonstrates the best performance for the SLM. iii) **Appendix D.5** evaluates the communication costs and running times, where FedTDP shows the superior performance in terms of efficiency compared to other baselines. iv) **Appendix D.6** evaluates the effect of FedTDP's hyperparameter, where the suggested value of $m$ is 25% or less.

## 6 Conclusion and Limitations

This paper introduces FedTDP, a privacy-preserving, unified framework for trajectory data preparation. It proposes a trajectory privacy autoencoder to protect data while maintaining spatio-temporal correlations, a trajectory knowledge enhancer to develop TDP-oriented LLMs, and parallel optimization to boost efficiency. Experiments on 6 datasets and 10 TDP tasks validate its superior effectiveness, efficiency, and robustness. There are some limitations of our work. Since this study mainly focuses on trajectory data preparation tasks, extending the framework to support more trajectory analysis tasks, such as clustering and pattern mining, remains an open opportunity. In addition, improving the interpretability of using LLMs poses an important challenge for future research.

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

587 # Appendix

588

# Appendix

In the subsequent sections, we present supplementary materials to provide more details of this paper, offering deeper insights and additional technical details for readers seeking further clarification. The appendix is organized as follows.

In **Section A**, we present a systematic review of related work to help readers understand the key development in areas relevant to this paper, including (i) the latest trajectory data preparation methods, (ii) applications of large language models in other data preparation, and (iii) recent advancements in large language models for spatio-temporal data analysis.

In **Section B**, we summary the preliminary of notations and trajectory data preparation tasks for better understanding our work, including (i) the frequently used notations and (ii) detailed description of trajectory data preparation tasks.

In **Section C**, we provide the additional methodology details to support the analysis shown in the main body of this paper, including (i) complete theoretical proofs of Theorems 1 and 2, (ii) practical examples of trajectory data preparation tasks using the trajectory prompt engineering of the proposed trajectory knowledge enhancer, and (iii) the training process of FedTDP.

In **Section D**, we describe the extensive experimental details to provide more information about experimental settings and further demonstrate the superiority performance of the proposed FedTDP framework, including (i) datasets description, (ii) compared baselines introduction, and (iii) the details experimental results of model generalization, model base, efficiency, and parameter sensitivity studies.

## A  Related Work

**Trajectory Data Preparation.** Numerous Trajectory Data Preparation (TDP) methods have been proposed to improve the quality of trajectory data for trajectory data preparation tasks. For the anomalous detection task, ATROM [22] addresses the critical challenge of anomaly recognition in open-world scenarios through the development of a probabilistic metric learning model, which significantly improves the accuracy of anomaly detection in complex environments. For the trajectory imputation task, Kamel [47] proposes a scalable architecture that incorporates additional real trajectory points to predict the missing trajectory data, improving the accuracy of trajectory imputation. For the map matching task, GraphMM [43] leverages the graphical structure using a graph neural network to effectively model the topology of road network and trajectory features, improving the accuracy of map matching. For the trajectory-user link task, AttnTUL [7] introduces a hierarchical spatio-temporal attention neural network, which co-encodes local trajectory transition patterns and global spatial dependencies to establish links between trajectories and users more accurately. For the travel mode identification task, Estimator [27] proposes an effective and scalable framework that partitions the traffic space into disjoint spatial regions based on traffic conditions, improving the accuracy of travel mode identification. For the trajectory simplification task, S3 [18] presents a lightweight framework consisting of two chained sequence-to-sequence modules, which is integrated within a graph neural architecture, improving the accuracy and efficiency of trajectory simplification. For the trajectory recovery task, LightTR [44] presents an efficient framework using a local trajectory embedding module, robust feature extraction capabilities while significantly reducing computational overhead. However, none of these studies have considered data privacy constraints. They typically assume that the trajectory data collection is centralized, which introduces a significant risk of privacy leakage, especially in federated learning environments. In addition, all of them are single-task methods. When handling multiple TDP tasks, different models need to be trained for each specific task. It not only demands substantial time and computational resources but also results in poor model generalization ability. ***In contrast, we aim to propose a privacy-preserving and unified framework to support various trajectory data preparation tasks while safeguarding trajectory data privacy.***

**Large Language Models for Other Data Preparation.** A few works on table data preparation using Large Language Models (LLMs) have been proposed recently. For instance, MELD [72] introduces a general solver for low-resource table data preparation, leveraging a mixture-of-experts architecture to support merging and augmentation of domain-specific experts trained on limited annotated examples. Similarly, TableGPT [35] presents a table-tuning paradigm, where LLMs are fine-tuned using various table tasks synthesized from real tables to enhance the model's ability to understand and process

table-related tasks. Additionally, [48] explores the performance of LLMs for table data preparation tasks, which evaluates their performance on five data cleaning and integration tasks through prompt-based methods. However, these works are specifically tailored for table data preparation and are not directly applicable to trajectory data preparation tasks. They lack the necessary understanding of trajectory data and do not account for the spatio-temporal characteristics and complexity of trajectory data preparation tasks, making them unsuitable for such applications. ***In contrast, we aim to develop a TDP-oriented LLM to effectively support various trajectory data preparation tasks.***

**Large Language Models for Spatio-Temporal Data Analysis.** There are a few spatio-temporal LLMs proposed [38, 82, 36], which have achieved superior performance in spatio-temporal downstream applications. Specifically, UrbanGPT [37] integrates a spatio-temporal dependency encoder with a command adjustment paradigm to enhance the LLMs' understanding of complex temporal and spatial interdependencies. Besides, UniST [77] develops a general-purpose model for urban spatio-temporal prediction through diverse data utilization, effective pre-training, and knowledge-guided prompts. In addition, PromptGAT [12] employs prompt-based grounded action transformations to analyze system dynamics by leveraging reasoning capabilities of large language models to understand environmental impacts on traffic patterns. However, none of them have considered trajectory data quality. If the quality of trajectory data is extremely poor, the performance of spatio-temporal large language models in downstream tasks will not be satisfactory either. ***In contrast, we aim to explore the powerful capabilities of large language models for trajectory data preparation to enhance the quality of trajectory data.***

# B    Notation and Trajectory Data Preparation Task

**Notation and Description.** We first present the frequently used notations and descriptions in this paper, as listed in Table 2.

Table 2: Notation and description

| Notation | Description |
|---|---|
| $p$ | A spatio-temporal point consisting of location and time $\langle l, t \rangle$ |
| $T$ | A trajectory consisting of multiple spatio-temporal points $\{p_1, p_2, \dots\}$ |
| $ST$ | A sub-trajectory of the trajectory $T$ |
| $S$ | A data silo that represents a region |
| $C$ | A client that represents a region |
| $D$ | The trajectory database in a client $C$ |
| $\mathcal{C}$ | A set of clients $\{C_1, C_2, \dots\}$ |
| $\mathcal{D}$ | A set of trajectory datasets $\{D_1, D_2, \dots\}$ |
| $\theta_{LLM}, \theta_{SLM}$ | The large language model and small language model |

**Trajectory Data Preparation Task.** Besides, we summary the supported trajectory data preparation tasks of the proposed FedTDP in this paper, as shown in Table 3, with the rough processing of each task shown in Fig. 9.

Table 3: Trajectory data preparation task

| Category | Task | Description |
|---|---|---|
| Data Cleaning | Anomaly Detection (AD) | Detect anomalous trajectory |
| | Trajectory Imputation (TI) | Predict missing points |
| | Noise Filtering (NF) | Filter point noise |
| | Stay Point Detection (SPD) | Identify stationary points |
| Data Matching | Map Matching (MM) | Align a trajectory to road network |
| | Trajectory-User Linking (TUL) | Associate trajectories with users |
| Data Annotation | Travel Mode Identification (TMI) | Identify transportation mode |
| Data Reduction | Trajectory Simplification (TSim) | Remove number of points |
| | Trajectory Segmentation (TSeg) | Divide a trajectory to segments |
| Data Augmentation | Trajectory Recovery (TR) | Recovery complete trajectory |

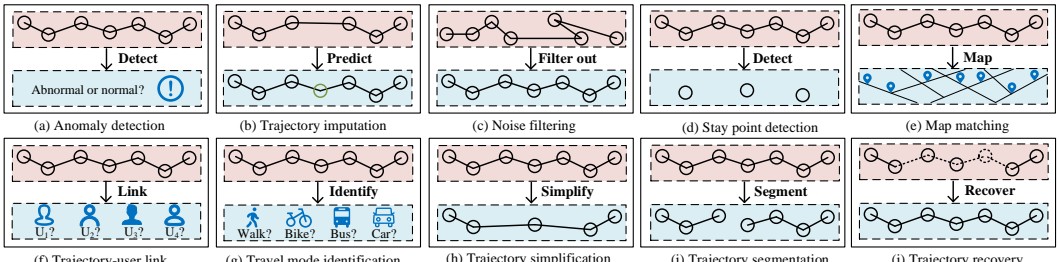

Figure 9: The supported trajectory data preparation tasks

## C  Additional Methodology Details

### C.1  Proof of Theorem 1

We provide the complete theoretical proof of Theorem 1 proposed in this paper that provides the correctness of the trajectory privacy autoencoder model aggregation, as detailed below.

*Proof.* According to Eq. 2, we can get the result of aggregating masked parameter blocks $\{\tilde{P}_1^{(k)}, \tilde{P}_2^{(k)}, \dots, \tilde{P}_{|\mathcal{C}|}^{(k)}\}$ for all clients, as formally shown below:

$$\sum_{i=1}^{|\mathcal{C}|} \tilde{P}_i^{(k)} = \sum_{i=1}^{|\mathcal{C}|} P_i^{(k)} + \sum_{i=1}^{|\mathcal{C}|} \sum_{j=1 \& j \neq i}^{|\mathcal{C}|} a_{i,j} * sk_{i,j} \tag{11}$$

Here, $sk_{i,j} = sk_{j,i}$ and $a_{i,j} = -a_{j,i}$, thus we can get the result formally shown below:

$$\sum_{i=1}^{|\mathcal{C}|} \sum_{j=1 \& j \neq i}^{|\mathcal{C}|} a_{i,j} * sk_{i,j} = \sum_{i=j+1}^{|\mathcal{C}|} \sum_{j=1}^{|\mathcal{C}|} (a_{i,j} * sk_{i,j} + a_{j,i} * sk_{j,i}) = 0 \tag{12}$$

The aggregation $\tilde{P}^{(k)}$ of the masked parameter block is formally shown below:

$$\sum_{i=1}^{|\mathcal{C}|} \tilde{P}_i^{(k)} = \sum_{i=1}^{|\mathcal{C}|} P_i^{(k)} + \sum_{i=1}^{|\mathcal{C}|} \sum_{j=1 \& j \neq i}^{|\mathcal{C}|} a_{i,j} * sk_{i,j} = \sum_{i=1}^{|\mathcal{C}|} P_i^{(k)} \tag{13}$$

$\square$

### C.2  Examples of Trajectory Prompt Engineering

For better understanding the trajectory prompt engineering of the proposed trajectory knowledge enhancer, we show practical examples of noise filtering and travel mode identification tasks using the trajectory prompt engineering in the small language model of the client.

As shown in Fig. 10, ***Task*** shows the task name with its description listed in Section 3. Besides, ***Data*** uses the raw trajectory data in clients, consisting of spatio-temporal points i.e., $T = \{p_1, p_2, \dots\}$. Additionally, ***Information*** includes optional road network and weather data. Specifically, *sunny* and 15 represent the weather tokens during the trajectory, *id* denotes the segment ID, *start-lon* and *start-lat* indicate the segment's longitude and latitude at the starting point, and *stop-lon* and *stop-lat* denote the segment's longitude and latitude at the stopping point. Specifically, we utilize the weather data for these tasks and the road network for the travel mode identification task. Moreover, ***Format*** refers to the expected output data of the task. We anticipate the outputs of noise filtering and travel mode identification to be the trajectory data and travel mode, respectively. Finally, the model's response should return the expected results in ***Format***. Note that, in the server ***Data*** uses embeddings uploaded by clients, consisting of encoded spatio-temporal points i.e., $E = \{e_1, e_2, \dots\}$.

| T-3: Noise Filtering | T-7: Travel Mode Identification |
|---|---|
| **Task:** it is a noise filtering task. It targets identifying and removing irrelevant spatio-temporal points that deviate from a trajectory. These noisy points can result from GPS inaccuracies, signal interference, or sensor malfunctions. | **Task:** it is a travel mode identification task. It aims to identify the travel mode of a trajectory based on the moving pattern of data. The travel mode is usually the walk, bike, bus, or car. |
| **Data:** the trajectory data consisting of spatio-temporal points is {*(lon₁, lat₁, t₁), (lon₂, lat₂, t₂), ...*}. | **Data:** the trajectory data consisting of spatio-temporal points is {*(lon₁, lat₁, t₁), (lon₂, lat₂, t₂), ...*}. |
| **Information:** the weather is *sunny* with an average temperature of *15*. | **Information:** It is *sunny* with an average temperature of *15*. The road network of the map is {*(id₁, start-lon₁, start-lat₁, stop-lon₁, stop-lat₁), (id₂, start-lon₂, start-lat₂, stop-lon₂, stop-lat₂), ...*}. |
| **Format:** the output should be the trajectory data. | **Format:** the output should be the travel mode. |
| **Model response:** the result of noise filtering is {(lon₁, lat₁, time₁), (lon₂, lat₂, time₂), ...}. | **Model response:** the travel mode is car. |

Figure 10: The example of trajectory prompt engineering in clients

## C.3 Proof of Theorem 2

We provide the complete theoretical proof of Theorem 2 proposed in this paper that determines the probability of parameters that need to be trained in the LoRA sparse-tuning of the proposed trajectory knowledge enhancer, as detailed below.

*Proof.* To derive the probability that each layer is selected when $N_m$ layers are chosen for the next round of training, we first calculate the probability that each layer is selected in the first time, which is equal to the ratio of the layer, as formally shown below:

$$Pr_1^{(r+1)}(L_i, N_m) = R^{(r)}(L_i) \tag{14}$$

Then, we calculate the probability that the layer is not selected in the first time and is selected in the second time, as formally shown below:

$$Pr_2^{(r+1)}(L_i, N_m) = \sum_{j_1=1}^{N} \frac{R^{(r)}(L_i) * R^{(r)}(L_{j_1})}{1 - R^{(r)}(L_{j_1})}, j_1 \neq i \tag{15}$$

Then, we calculate the probability that the layer is not selected in the first two times and is selected in the third time, as formally shown below:

$$Pr_3^{(r+1)}(L_i, N_m) = \sum_{j_1=1}^{N} \sum_{j_2=1}^{N} \frac{R^{(r)}(L_i) * R^{(r)}(L_{j_1}) * R^{(r)}(L_{j_2})}{1 - R^{(r)}(L_{j_1}) - R^{(r)}(L_{j_2})}, \tag{16}$$

where $j_1 \neq j_2 \neq i$. Based on the above inductive steps, we can calculate the probability that the layer is not selected in the first $N_m - 1$ times and is selected in the $N_m$ time by inductive reasoning, as formally shown below:

$$Pr_{N_m}^{(r+1)}(L_i, N_m) = \sum_{j_1=1}^{N} \cdots \sum_{j_{N_m}=1}^{N} \frac{R^{(r)}(L_i) * R^{(r)}(L_{j_1}) * \ldots * R^{(r)}(L_{j_{N_m}})}{1 - R^{(r)}(L_{j_1}) - \ldots - R^{(r)}(L_{j_{N_m}})}, \tag{17}$$

where $j_1 \neq \ldots \neq j_{N_m} \neq i$. Thus, the probability that layer $L_i$ is selected to train in the next round $r + 1$ can be calculated as follows:

$$Pr^{(r+1)}(L_i, N_m) = \sum_{j=1}^{N_m} Pr_j^{(r+1)}(L_i, N_m) = R^{(r)}(L_i) + \sum_{j_1=1}^{N} \frac{R^{(r)}(L_i) * R^{(r)}(L_{j_1})}{1 - R^{(r)}(L_{j_1})} + \ldots +$$
$$\sum_{j_1=1}^{N} \cdots \sum_{j_{N_m}=1}^{N} \frac{R^{(r)}(L_i) * R^{(r)}(L_{j_1}) * \ldots * R^{(r)}(L_{j_{N_m}})}{1 - R^{(r)}(L_{j_1}) - \ldots - R^{(r)}(L_{j_{N_m}})}, \tag{18}$$

where $j_1 \neq \ldots \neq j_{N_m} \neq i$. $\square$

 **C.4   Multi-Task Training**

Due to the diverse range of trajectory data preparation tasks that need to be addressed, we propose a multi-task training strategy to enhance the model's learning and generalization capabilities. Specifically, we prepare a trajectory dataset applicable to most trajectory data preparation tasks and construct labels for each task. During the training phase, we execute multiple trajectory data preparation tasks on the same trajectory data input, calculate the loss for each task, and jointly optimize the model formulaically shown below:

$$\mathcal{L} = \mathcal{L}_{T\text{-}1} + \mathcal{L}_{T\text{-}2} + \ldots + \mathcal{L}_{T\text{-}10} \,, \tag{19}$$

where $\mathcal{L}_{T\text{-}i}$ is the loss of trajectory data preparation task T-$i$ listed in the Section 3. Note that the proposed FedTDP framework can be easily extended to support other trajectory data preparation tasks benefiting from its modular architecture, decoupled data processing pipeline, and variable model base.

**Training Algorithm.** For convenient method reproduction, we provide a detailed training process of the entire FedTDP framework, which can be divided into the server and client, as shown in Algorithms 1 and 2.

---

**Algorithm 1:** The training on the server

**Input:** the number of training rounds *TR*

1 **for** *round $r = 0, \ldots, TR - 1$* **do**
2    $f \leftarrow$ IsFrozen($r$) // Get the frozen status of this round.;
3    **if** $f == False$ **then**
4      $E \leftarrow$ Get($\mathcal{C}$) // Get the embeddings data from clients.;
5      $E \leftarrow$ Connect($E$) // Connect into a complete embeddings.;
6    **else**
7      $E \leftarrow$ GetFrozenData($r - 1$) // Get the frozen data.;
8    $prompt \leftarrow$ TKE($E$) // Construct the prompt of the embeddings.;
9    $o \leftarrow \theta_{\text{LLM}}(prompt)$ // Input the prompt data to the LLM.;
10    $o \leftarrow$ Split($o$) // Split the output of the LLM.;
11    **if** $f == False$ **then**
12      **for** *client number $i = 0, \ldots, |\mathcal{C}| - 1$* **do**
13        Send($o_i, C_i$) // Send split results to respective clients.;

---

**Algorithm 2:** The training on the client

**Input:** the number of training rounds *TR* and server $s$

1 **for** *round $r = 0, \ldots, TR - 1$* **do**
2    $D \leftarrow$ GetData() // Get the local trajectory data.;
3    $f \leftarrow$ IsFrozen($r$) // Get the frozen status of this round.;
4    **if** $f == False$ **then**
5      $E \leftarrow$ Enc($D$) // Encode the trajectory into embeddings.;
6      Send($E, s$) // Send the embeddings data to the server.;
7    $prompt \leftarrow$ TKE($D$) // Construct the prompt of the data.;
8    $o' \leftarrow \theta_{\text{SLM}}(prompt)$ // Input the prompt data to the SLM.;
9    **if** $f == False$ **then**
10      $o \leftarrow$ Get($s$) // Get the result from the server.;
11      $o \leftarrow$ Dec($o$) // Decode the server's result.;
12    **else**
13      $o \leftarrow$ GetFrozenData($r - 1$) // Get the frozen data.;
14    $result \leftarrow$ TKE($o', o$) // Compute the distillation result.;

---

In the server (i.e., Algorithm 1), the input is the number of training rounds (line 1). For each training round $r$, it begins to get the frozen state $f$ (lines 2–3). If $f$ is not frozen, the server gets the trajectory

embeddings $E$ from clients $\mathcal{C}$ and connects them, or it gets local $E$ frozen in the last training round $r - 1$ (lines 4–9). Then, the server uses TKE to construct the TDP prompt for the LLM and get the output $o$ (lines 10–11). Finally, the server splits it into several parts and sends them to respective clients if $f$ is not frozen (lines 12–18).

In the client (i.e., Algorithm 2), the input are the number of training rounds and the server (line 1). For each training round $r$, it begins to get the trajectory data $D$ and frozen state $f$ (lines 2–4). If $f$ is not frozen, clients encode $D$ into embeddings and send them to the server (lines 5–8). Then, clients use TKE to construct the TDP prompt for the SLM and get the output $o^{'}$ (lines 9–10). If $f$ is not frozen, clients get the LLM's output $o$ from the server and decode it, or it gets local $o$ frozen in the last round $r - 1$ (lines 11–16). Finally, clients use TKE to compute the final result between $o^{'}$ and $o$ (lines 17–18).

**Complexity Analyses.** We also give complexity analyses for Algorithms 1 and 2. Specifically, given the number of trajectory embeddings data $|E|$ from all clients, the complexity of Algorithm 1 is $O(|E| * TR * MC)$, where $MC$ is the model complexity of the LLM. Given the number of trajectories $|D|$ in the client, the complexity of Algorithm 2 is $O(|D| * TR * MC^{'})$, where $MC^{'}$ is the model complexity of the SLM.

## C.5 Theoretical Privacy Analysis

The privacy protection mechanism of the proposed FedTDP framework is built upon the Trajectory Privacy Autoencoder (TPA), which protects trajectory data privacy while maintaining spatio-temporal correlations. Besides, it develops a decentralized aggregation approach based on secret sharing [58] that ensures the parameters of the TPA model remain confidential against trajectory data recovery or inference through embedding [61, 8, 30] and gradient [67, 84, 87] inversion attacks. To rigorously analyze the privacy-preserving capability of TPA, we first define the threat model as follows.

**Threat Model**. Following prior works [81, 63, 83] in federated learning, we assume the server acts as a semi-honest adversary who will honestly execute required operations (e.g., aggregation) but also remains curious about the private client data. In the F-TDP problem, the server seeks to reconstruct clients' raw trajectory data using adversary's knowledge, which includes the client model architecture, including the client model architecture, the privacy-preserving mechanism, and the encoded embeddings uploaded by clients.

Based on this, we use mutual information [32] to quantify the upper bound of privacy leakage, i.e., $I(T; E)$, which measures the information about original trajectory data $T$ that can be inferred from encoded embeddings $E$ transmitted to the server, as shown below:

$$I(T; E) = H(T) - H(T|E), \tag{20}$$

where $H(\cdot)$ denotes the entropy. Since $E$ is derived from $T$ through the encoder $\theta_{Enc}$, the conditional entropy $H(T|E)$ can be decomposed as:

$$H(T|E) = \mathbb{E}_{\theta_{Enc} \sim \mathcal{P}_{\Theta}}[H(T|E, \theta_{Enc})] + H(\theta_{Enc}|E), \tag{21}$$

where $\theta_{Enc}$ is drawn from a prior distribution $\mathcal{P}_{\Theta}$ and $\Theta$ is the parameter space. Besides, both $H(T|E, \theta_{Enc})$ and $H(\theta_{Enc}|E)$ are large because $\theta_{Enc}$ is private and inaccessible to the server. Consequently, the conditional entropy $H(T|E)$ remains high, leading to minimal privacy leakage $I(T; E)$. Furthermore, leveraging Bayes' theorem [3] and Fano's inequality [19], the probability $P_e$ of the attacker recovering $T$ incorrectly satisfies:

$$H(P_e) + P_e log|\mathcal{T}| \geq H(T|E), \tag{22}$$

where $\mathcal{T}$ denotes the trajectory data space. The large $H(T|E)$ results in a correspondingly high $P_e$, indicating a low likelihood of successful reconstruction of the original trajectory data, which further underscores the effectiveness of TPA in protecting trajectory privacy.

# D Experimental Details

## D.1 Datasets

We evaluate the proposed FedTDP framework using 6 datasets, including GeoLife [86], Porto [51], T-Drive [76], Tencent [43], Gowalla [11], and SHL [60], with their statistics shown in Table 4, as detailed below.

Table 4: The statistics of dataset

| Dataset | # trajectories | # points | Quality Issue | Task |
|---------|---------------|----------|---------------|------|
| GeoLife | 182 | 24,876,978 | Positional inaccuracies, data noise, and lower precision | AD, TI, MM, TUL, TMI, TSim, and TR |
| Porto | 442 | 83,409,386 | Anomalies and missing data | AD and TI |
| T-Drive | 10,336 | 17,662,984 | Noisy and incomplete points | NF, TR, SPD, and TSim |
| Tencent | 40,966 | 1,610,216 | Inaccurate points | MM |
| Gowalla | 107,092 | 6,442,890 | Sparse and non-continuous data | TUL |
| SHL | 3 | 109,390 | Duplicate records | TSeg and TMI |

- **GeoLife [86].** It collected 182 trajectories with 24,876,978 spatio-temporal points, used for the training tasks, including Anomaly Detection (AD), Trajectory Imputation (TI), Map Matching (MM), Trajectory-User Link (TUL), Travel Mode Identification (TMI), Trajectory Simplification (TSim), and Trajectory Recovery (TR) tasks. It contains various quality issues such as positional inaccuracies, data noise, and lower precision due to irregular sampling intervals and sensor limitations, which make it suitable for various trajectory data preparation tasks.

- **Porto [51].** It collected 442 trajectories with 83,409,386 spatio-temporal points, which contains quality issues such as anomalies and missing data, used for AD and TI testing tasks.

- **T-Drive [76].** It collected 10,336 trajectories with 17,662,984 spatio-temporal points, which contains quality issues such as noisy and incomplete points, used for NF, TR, SPD, and TSim testing tasks.

- **Tencent [43].** It collected 40,966 trajectories with 1,610,216 spatio-temporal points, which contains quality issues such as inaccurate points due to the low sampling rate, used for the MM testing task.

- **Gowalla [11].** It collected 107,092 trajectories with 6,442,890 spatio-temporal points, which contains quality issues such as sparse and non-continuous data, used for the TUL testing task.

- **SHL [60].** It collected 3 trajectories with 109,390 spatio-temporal points, which contain quality issues such as duplicate records, used for TSeg and TMI testing tasks.

## D.2 Baselines

We compare the proposed FedTDP framework with state-of-the-art baselines, as shown in Table 5.

Table 5: The compared baselines

| Category | Task | Method | Year |
|----------|------|--------|------|
| | Anomaly Detection | ATROM | 2023 |
| | Trajectory Imputation | Kamel | 2023 |
| | Map Matching | GraphMM | 2024 |
| **S-TDP** | Trajectory-User Linking | AttnTUL | 2024 |
| | Travel Mode Identification | Estimator | 2024 |
| | Trajectory Simplification | S3 | 2023 |
| | Trajectory Recovery | LightTR | 2024 |
| **Large Language Models for Table Data Preparation** | | FM4DP | 2022 |
| | | MELD | 2024 |
| | All tasks for evaluation | TableGPT | 2024 |
| | | PromptGAT | 2024 |
| **Large Language Models for Spatio-Temporal Data Analysis** | | UniST | 2024 |
| | | UrbanGPT | 2024 |

First, we compare FedTDP with various Trajectory Data Preparation (TDP) methods in a single TDP task (referred to S-TDP), including ATROM [22] for the AD task, Kamel [47] for the TI task, GraphMM [43] for the MM task, AttnTUL [7] for the TUL task, Estimator [27] for the TMI task, S3 [18] for the TSeg task, and LightTR [44] for the TR task, as detailed below.

- **ATROM [22].** It solves the anomaly detection task in open-world scenarios and introduces a new probabilistic metric learning model.

- **Kamel [47].** It proposes a scalable system that inserts additional real trajectory points to improve the accuracy of the trajectory imputation task.

- **GraphMM [43].** It develops the graphical nature of the map matching task to exploit the road network and trajectory graphical topology.
- **AttnTUL [7].** It proposes a hierarchical trajectory attention neural network for co-encoding local trajectory transition patterns and global spatial dependencies to solve the trajectory-user link task.
- **Estimator [27].** It partitions the entire traffic space into disjoint spatial regions based on the traffic conditions for the travel mode identification task.
- **S3 [18].** It presents a lightweight trajectory segmentation task framework to augment the trajectory representation paradigm with geo-semantics.
- **LightTR [44].** It develops a local trajectory embedding module that provides higher computational efficiency for the trajectory recovery task.

Besides, we compare FedTDP with three methods using Large Language Models (LLMs) for table data preparation, including FM4DP [48], MELD [72], and TableGPT [35], as detailed below.

- **FM4DP [48].** It helps LLMs understand table DP tasks, which uses 5 data cleaning and integration table DP tasks as prompt tasks and evaluates the performance of LLMs on these tasks.
- **MELD [72].** It employs the mixture-of-experts architecture to support the merging and augmentation of experts trained on the domain-specific experts trained on limited annotated examples.
- **TableGPT [35].** It proposes a table-tuning paradigm using various table tasks synthesized from real tables as the training data to help LLMs understand the table data and perform table tasks.

Moreover, we compare FedTDP with three LLM-based models for spatio-temporal data analysis, including PromptGAT [12], UniST [77], and UrbanGPT [37], as detailed below.

- **PromptGAT [12].** It uses the LLM to analyze system dynamics, leveraging the context and spatio-temporal data to understand how weather, traffic, and road conditions affect traffic dynamics.
- **UniST [77].** It proposes a general-purpose model, which is designed for urban spatio-temporal prediction in various urban scenarios to capture the complex spatio-temporal relationship.
- **UrbanGPT [37].** It integrates a spatio-temporal dependency encoder with a command adjustment paradigm, which enables LLMs to understand complex spatio-temporal interdependencies.

### D.3 Model Generalization Study

To evaluate the proposed FedTDP framework generalization in different numbers of training tasks, we systematically remove the training task from back to front based on their order in Table 1. As illustrated in Fig. 11, the results indicate that the performance of FedTDP across various tasks declines as the number of training tasks decreases. This decline is primarily attributed to the reduced acquisition of TDP knowledge, which adversely affects generalization. Notably, when the number of tasks is reduced to one (i.e., training FedTDP solely on the anomaly detection task using GeoLife), the performance also falls below that of S-TDP in Fig. 5

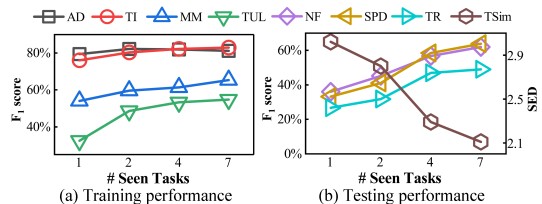

Figure 11: Scalability study

### D.4 Model Base Study

To evaluate the impact of various model bases on the proposed FedTDP framework, we choose widely used model bases for the LLM (Llama-8B [64], GPT3-7B [5], and Qwen-7B [2]) and SLM (GPT3-Small-125M [5], GPT2-Small-137M [52], and T5-Small-60M [53]). Besides, to evaluate the impact of different LLMs on FedTDP, we use GPT3-Small as the client's SLM while we use Llama as the server's LLM to evaluate the impact of different SLMs on FedTDP. The results are shown in Fig. 12. As observed, Llama achieves optimal performance in most TPD tasks for the server's LLM, followed by GPT3 and then Qwen. In contrast, GPT3-Small demonstrates the best performance for the client's SLM, succeeded by T5-Small and then GPT2-Small. Consequently, we adopt Llama and GPT3-Small as the default server's LLM and client's SLM in other experiments, respectively.

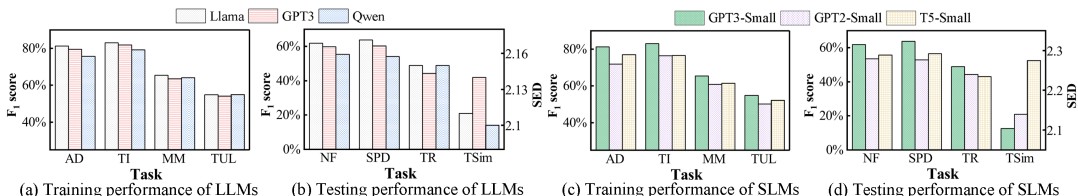

(a) Training performance of LLMs     (b) Testing performance of LLMs     (c) Training performance of SLMs     (d) Testing performance of SLMs

Figure 12: Large language model and small language model base study

## D.5 Efficiency Study

Fig. 13 shows the communication costs (in GB) and running times (in hours) of various methods across all TDP tasks. During the training process, the proposed framework incurs the largest communication size because it must transfer embeddings and model parameters, whereas LLM-based methods (i.e., LLMs for table table data preparation and spatio-temporal LLMs) transmit all perturbed data, generated through differential privacy, to the server in the first round of

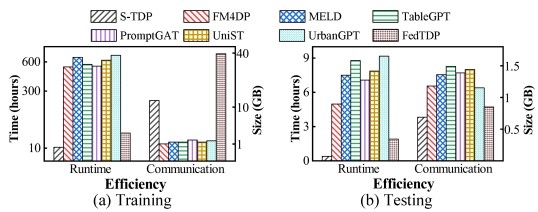

(a) Training          (b) Testing

Figure 13: Efficiency study

training and do not require data transmission in the following rounds of training. Furthermore, the runtime of FedTDP is reduced by a factor of **11.3 to 14.2** compared to other LLM-based methods, demonstrating its efficiency in the F-TDP context. In the testing phase, the communication size of FedTDP is nearly identical to that of S-TDP and **1.4 to 1.8** times less than that of other LLM-based methods, which require transferring all data to the server, whereas FedTDP only transmits the data necessary for cross-client TDP. Additionally, the runtime of FedTDP is **2.6 to 4.8** times lower than that of other LLM-based methods, further underscoring its efficiency.

## D.6 Parameter Sensitivity Study

We evaluate the effects of hyperparameters of the proposed FedTDP framework (i.e., the training layers ratio $m$ of LoRA sparse-tuning in the TKE module), as shown in Fig. 14, where we change the $m$ from 25% to 100%. We can observe that as $m$ increases, the performance of FedTDP improves slightly. However, this improvement comes at the cost of increased training time and communication size, as the number of parameters that need to be trained and transmitted also rises when $m$ is increased. There-

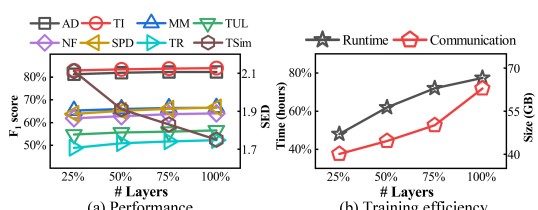

(a) Performance          (b) Training efficiency

Figure 14: Parameter sensitive study

fore, the suggested value of $m$ is 25% or less, as long as the model performance with the value of $m$ is acceptable.

