# OpenReview forum: "A Privacy-Preserving and Unified Federated Learning Framework for Trajectory Data Preparation"
_NeurIPS.cc/2025/Conference — Submitted to NeurIPS 2025_

### Official Review · Reviewer_P2Gk · 2025-06-29

**Clarity:** 2
**Significance:** 2
**Originality:** 2
**Rating:** 3
**Confidence:** 4

**Summary:**

This paper proposes FedTDP, a privacy-preserving and unified federated learning framework for trajectory data preparation, which effectively combines the strengths of SLMs and LLMs. The framework is designed to address two key challenges in real-world trajectory data applications: the protection of user privacy in decentralized environments and the need for accurate, efficient trajectory data processing. It introduces the Trajectory Privacy Autoencoder to ensure that sensitive user location information is protected during training, and the Trajectory Knowledge Enhancer to improve the model’s understanding of spatio-temporal dependencies through prompt-based knowledge transfer, LoRA-based parameter-efficient tuning, and bidirectional knowledge distillation between SLMs and LLMs.
The contributions of this work are as follows: (1) it is the first framework that integrates federated learning with large language models for trajectory data preparation; (2) it proposes a privacy-preserving autoencoder that supports global training without sharing raw trajectory data; (3) it introduces a novel trajectory-specific prompting and tuning mechanism to enhance LLM adaptability to TDP tasks.

**Questions:**

Q1:In the Trajectory Knowledge Enhancer module, the authors mention that "Information" (e.g., road network, weather) is an optional context to enhance model performance on TDP tasks. Could the authors clarify how the model determines whether to include this additional context and which type of information to use for a given task? Is this decision task-specific, manually configured, or automatically inferred during prompt construction? It would be helpful to provide either a decision rule, algorithmic selection process, or ablation comparison showing the impact of using different types of Information (or none).

Q2: The Trajectory Offsite-Tuning section introduces an adapter-based strategy where the adapter A is trained and aggregated.What is the specific relationship between adapter A and the N layers of the base model? Are adapters inserted in all N layers but only a subset AAA of them are aggregated, or are only A layers fine-tuned and aggregated?
Additionally, how is the adapter structure shared across different base models, especially across families of SLMs and LLMs? Providing more details on the layer-wise configuration and compatibility requirements for adapter sharing across model families would clarify scalability and extensibility.

**Ethical Concerns:**

["NO or VERY MINOR ethics concerns only"]

**Final Justification:**

I have carefully red rebuttals and considered all reviews. However, the technical contribution is not strong. As a result, my rating remains borderline reject.

**Limitations:**

See weaknesses.

**Quality:**

2

**Strengths And Weaknesses:**

Strengths:

S1.The paper is generally well-written and logically structured, with explanations of core modules that facilitate understanding, although some parts may be difficult for first-time readers.

S2. It introduces a novel problem setting by defining and studying federated trajectory data preparation (F-TDP) for the first time, showing potential impact through its integration of federated learning with trajectory data preparation.

S3. The framework demonstrates notable originality by combining federated learning, prompt-based LLM fine-tuning, and privacy-preserving encoders for trajectory data, using bidirectional knowledge distillation between SLMs and LLMs along with LoRA-based sparse tuning in a split-learning setting, which has not been explored before.

Weaknesses:

W1.Figure 2 lacks clarity in illustrating the interaction between local and cross-client TDP workflows, and the depiction of split learning and alternating optimization could be more intuitive, suggesting the need to visually separate the two pipelines and improve step labeling for readability.

W2.Although F-TDP is proposed as a novel task, its necessity and motivation could be more strongly justified since it is unclear why this specific formulation is essential or how it fundamentally differs from traditional federated learning tasks, and related work does not discuss prior FL applications to TDP while baseline comparisons are imbalanced due to the lack of privacy-preserving designs.

W3.While the integration is innovative, the individual components are known in literature and the paper could strengthen its originality by elaborating why this specific combination is essential for F-TDP and clarifying its distinctiveness, perhaps as a vertical federated problem.

---

> ### Author Rebuttal · Authors · 2025-07-31
>
> Thanks for all the valuable comments and questions. **We have addressed all the concerns as detailed below**.
> ```
> W1. Figure 2 lacks clarity in distinguishing local vs. cross-client TDP workflows and illustrating split learning and alternating optimization.
> ```
> We thank the reviewer for the suggestion. In the revised manuscript, we will improve Fig. 2 for clarify.
>
>
>
> ```
> W2. Although F-TDP is proposed as a novel task, its necessity and motivation could be more strongly justified since it is unclear why this specific formulation is essential or how it fundamentally differs from traditional federated learning tasks, and related work does not discuss prior FL applications to TDP while baseline comparisons are imbalanced due to the lack of privacy-preserving designs.
> ```
> Thank the reviewer for the thoughtful comment. We appreciate the opportunity to further clarify the **motivation and necessity** of our proposed **Federated Trajectory Data Preparation (F-TDP)** task. F-TDP is not merely a repackaging of federated learning (FL) applied to a new domain, but a **concrete and practically essential problem** that arises from real-world applications.
>
> - **Motivation 1: Legal and Structural Constraints on Trajectory Data**. Trajectory data is **highly sensitive**, as it can reveal fine-grained personal information (e.g., home/work locations, movement habits). Moreover, in practice: i) **trajectories are often split across regions** due to administrative boundaries or privacy regulations, as shown in **Figure 1** (page 2). ii) **centralized storage or cross-regional transmission is legally restricted** under regulations like **GDPR** and **CCPA**. These examples illustrate that **traditional FL settings**—which assume clients with full and independent data distributions—**do not fit the operational reality** of cross-region trajectory data management.
>
> - **Motivation 2: Generalization Gap in Existing TDP Methods**. Current trajectory data preparation (TDP) methods are i) **single-task**: Each task (e.g., imputation, map matching, anomaly detection) requires separate models and training processes. ii) **centralized**: Most TDP models assume centralized access to complete trajectories. This leads to: **high resource costs** (e.g., retraining for each task/domain) and **poor generalization** across tasks, datasets, and regions. In contrast, **F-TDP** is designed to support **unified multi-task modeling** via LLMs and **privacy-preserving learning** in decentralized environments with spatial partitioning.
>
> - **Regarding Baselines and Comparisons**. We acknowledge that most **existing TDP baselines do not consider privacy**, as this is an **emerging and under-explored area**. To address this, we:
> - Include **both LLM-based and non-LLM-based** baselines to evaluate generalizability.
>
> - Conduct **ablation studies** to isolate the impact of our privacy module (TPA), showing it incurs **minimal performance loss** while providing strong privacy protection (**Section 5.2, page 9**).
>
> *[1] Uber. Data governance framework set up by Uber. https://cionews.co.in/data-governance-framework-set-up-by-uber/. 2022.*
>
> *[2] Open Mobility Foundation. Mobility Data Specification - Jurisdiction Service. https://github.com/openmobilityfoundation/mobility-data-specification/tree/main/jurisdiction. 2019.*
>
> *[3] MobiSpaces. New data spaces for green mobility. https://mobispaces.eu/. 2022.*
>
>
> ```
> W3. While the integration is innovative, the individual components are known in literature and the paper could strengthen its originality by elaborating why this specific combination is essential for F-TDP and clarifying its distinctiveness, perhaps as a vertical federated problem.
> ```
> We appreciate the reviewer's comment and the opportunity to further clarify the **originality and necessity** of our proposed design. **FedTDP is not a simple composition of known techniques, but an integrated and novel framework—each component is purposefully crafted to address the below challenges that existing methods cannot effectively solve**. Below, we elaborate on how each module contributes **novel capabilities** that are **crucial and interdependent**.
>
> #### **(1) Challenge 1: Preserving Privacy in Cross-Client TDP Tasks**
>
> Traditional FL and DP-based mechanisms assume clients can locally complete tasks with minimal context from others. However, in **trajectory imputation, map matching, or anomaly detection**, **context from adjacent regions is critical**. Standard DP introduces random noise that disrupts **spatio-temporal continuity**, degrading downstream accuracy. In contrast, we propose the **Trajectory Privacy Autoencoder (TPA)**, which:
>
> - Encodes raw trajectories into **privacy-preserving spatio-temporal embeddings** that **retain essential semantics** (e.g., directionality, speed) across client boundaries.
> - Enables cross-client trajectory tasks **without exposing raw data**.
> - **Is theoretically proven** to be non-invertible when the encoder-decoder pair is private (**Appendix C.5, page 21**), providing **stronger protection** than traditional DP in federated applications.
>
> #### **(2) Challenge 2: Adapting LLMs to Heterogeneous and Task-Specific TDP Scenarios**
>
> Generic LLMs lack inductive bias for trajectory semantics, while existing spatio-temporal LLMs are task-specific (e.g., prediction) and unsuitable for general TDP preparation tasks. As shown in Figs. 6–7 (page 8), direct application of these LLMs yields poor results. In contrast, we introduce the **Trajectory Knowledge Enhancer (TKE)**, which integrates:
>
> - **Trajectory Prompt Engineering** to bridge spatio-temporal semantics and LLM inputs.
> - **Offsite-Tuning** to adapt LLMs to trajectory-specific patterns in a federated setting.
> - **Bidirectional Knowledge Learning** to incorporate both **TDP priors** and **trajectory features**.
>
> #### **(3) Challenge 3: Efficient Optimization Under Federated Constraints**
>
> In typical FL, clients either run heavy models locally or offload everything to the server. For F-TDP: i) LLMs are too large to deploy on clients, ii) centralizing embeddings from all clients overburdens the server, and iii) existing PEFT methods reduce training cost but not transmission or data handling costs. In contrast, we propose **Federated Parallel Optimization (FPO)**, which:
>
> - Deploys a **lightweight SLM on clients** for local adaptation.
> - Uses **frozen trajectory embeddings** to decouple LLM updates from client-side training.
> - Enables **parallel and asynchronous training** on clients and server, reducing overhead and improving convergence.
>
> #### **(4) Summary of Contribution/Novelty**
>
> While the individual elements may draw from established research, their **co-design and integration** in FedTDP is **original and essential** for solving the **practical, under-addressed F-TDP problem**. Specifically:
>
> - **TPA** ≠ DP or traditional autoencoders → enables cross-client privacy-aware context modeling.
> - **TKE** ≠ generic or spatio-temporal LLMs → enables multi-task trajectory preparation.
> - **FPO** ≠ standard FL optimizers → enables parallelism in heterogeneous federated LLM scenarios.
>
> Extensive experiments demonstrate that FedTDP **significantly outperforms 13 strong baselines with improvements from 4.84% to 45.22%**, validating both **its novelty and practical utility** in federated trajectory processing.
>
>
>
> ```
> Q1. In the Trajectory Knowledge Enhancer, "Information" (e.g., road network, weather) is optional. How does the model decide whether and which context to include, task-specifically, manually, or automatically?
> ```
> We thank the reviewer for this insightful question. The inclusion of the `"Information"` field in the Trajectory Knowledge Enhancer (TKE) is **manually configured** and **task-specific**, rather than determined automatically by the model.
>
> For example, the **road network** is essential included for the **Map Matching** task, where understanding road topology is critical. In contrast, **weather information** is optional and only used when it is expected to improve performance, such as in tasks where external conditions may influence mobility behavior. This manual configuration allows us to flexibly incorporate relevant context based on domain knowledge and the characteristics of each TDP task. We will clarify this design in the revision.
>
>
>
> ```
> Q2. In Trajectory Offsite-Tuning, the relationship between adapter A and the base model's N layers is unclear. Are adapters inserted in all layers but only some aggregated, or only A layers fine-tuned and aggregated? How is the adapter structure shared across SLM and LLM families?
> ```
> We thank the reviewer for this detailed question. We clarify the design of **Trajectory Offsite-Tuning** as follows:
>
> The adapter **A** is not inserted across the entire LLM. Rather, it refers to the **top-*k*** layers of the server's LLM. The remaining **N-k** layers constitute the fixed **foundation F**. During tuning, the **adapter A** is dispatched to the client and connected to the tail of the client's SLM. The client fine-tunes only **A** on local data, and the updated adapters are returned and aggregated on the server. This structure ensures that **only the decision-making layers** (adapter A) are optimized, enabling lightweight and targeted knowledge transfer without modifying the foundation model.
>
> To ensure compatibility across different model families, the only requirement is **hidden dimension alignment**: the output dimension of the SLM's final layer must match the input dimension of the LLM's adapter A. This flexible design allows offsite tuning to operate across heterogeneous models while enabling effective collaboration.

---

> > ### Comment · Reviewer_P2Gk · 2025-08-03
> > **rebuttal slightly improves**
> >
> > The authors has addressed some of my previous concerns. By considering with other reviewers' comments, I decide to remain my score unchanged.

---

> > > ### Author Response · Authors · 2025-08-04
> > > **Response to Reviewer P2Gk**
> > >
> > > We would like to sincerely thank the reviewer again for their thoughtful and timely feedback.
> > >
> > > **The main concerns previously raised—regarding the novelty of our proposed paradigm (W2, W3), the clarity of a figure (W1), and the need for more detailed explanations (Q1, Q2)—have been carefully addressed in our point-by-point responses.**
> > > In particular, we would like to note that the concerns about **novelty (W2, W3)** were not shared by the other reviewers, who generally recognized the **proposed framework as novel and impactful**. This aligns with our intention to position FedTDP as a distinct and innovative approach within the broader landscape of federated learning and spatio-temporal data preparation/pre-processing.
> > >
> > > As for the remaining concerns—such as clarity in figures (W1) and requests for additional details (Q1, Q2)—we believe these are **presentation-level or implementation-level issues**, and we have **carefully revised the paper to address each of them explicitly**.
> > >
> > > We greatly appreciate the reviewer’s engagement throughout the review process. Your feedback has been invaluable in helping us improve the clarity and completeness of our manuscript. We are committed to incorporating all suggestions into the final version.
> > >
> > > **If there are any remaining concerns or if any of our explanations were insufficient, we would be more than happy to provide further clarification or make additional revisions.** We welcome any further guidance that could help us strengthen the work.

---

> > > ### Author Response · Authors · 2025-08-06
> > > **Kind Reminder for Further Discussion**
> > >
> > > **We thank the reviewer for the constructive feedback**. However, we notice that the reviewer states that *"the authors have addressed some of my previous concerns".* Thus, we are committed to addressing any remaining issues and would be grateful for further clarification on specific points needing attention.
> > >
> > > The main concerns previously raised—regarding the novelty of our proposed paradigm (**W2, W3**), the clarity of a figure (**W1**), and the need for more detailed explanations (**Q1, Q2**)—have been carefully addressed in our point-by-point responses.
> > >
> > > Specifically, we note that concerns about novelty (W2, W3) were carefully clarified, as other reviewers explicitly recognized FedTDP’s innovation and impact. Regarding remaining issues—figure clarity (W1) and additional details (Q1, Q2)—we’ve addressed these presentation/implementation points thoroughly in the Rebuttal Response.
> > >
> > > Overall, we hope our clarifications and experimental results now demonstrate FedTDP’s contributions more clearly. **We sincerely appreciate the reviewer’s time and feedback and would be grateful for a reconsideration of the overall evaluation**. Once again, we thank the reviewer for the valuable time and effort invested in the review process.

---

> > > ### Author Response · Authors · 2025-08-07
> > > **Kind Reminder for Further Discussion**
> > >
> > > Dear Reviewer P2Gk,
> > >
> > > **Thank you very much for your engagement and for acknowledging that some of your concerns have been addressed**.
> > >
> > > As we mentioned in our response three days ago, we carefully addressed the key points you raised—regarding **the novelty of our paradigm**, **clarity of a figure**, and **the requested detailed explanations**. We also noted that other reviewers have recognized the contributions and impact of FedTDP.
> > >
> > > **We fully respect your decision to keep the score unchanged at this stage. At the same time, given that you have acknowledged some of the concerns have been addressed, we would sincerely appreciate it if you could kindly take this into consideration when finalizing your evaluation**.
> > >
> > > Regardless of the outcome, your constructive feedback has been invaluable to us, and we deeply appreciate your thoughtful review throughout the process.
> > >
> > > Thank you again for your time and consideration.
> > >
> > > Warm regards,
> > >
> > > The authors of Paper 10076

---

### Official Review · Reviewer_PDos · 2025-07-02

**Clarity:** 3
**Significance:** 3
**Originality:** 3
**Rating:** 4
**Confidence:** 3

**Summary:**

This paper proposes FedTDP, a novel framework for trajectory data preparation (TDP) in federated settings using LLMs. The framework introduces three main components:
- Trajectory Privacy AutoEncoder (TPA) to encode spatio-temporal data and protect privacy.
- Trajectory Knowledge Enhancer (TKE) for adapting LLMs/SLMs to TDP tasks via prompt engineering, offsite tuning, and LoRA sparse-tuning.
- Federated Parallel Optimization (FPO) to reduce training overhead using split learning and alternating optimization.
The authors evaluate FedTDP across 10 TDP tasks and 6 real-world datasets (seen/unseen), showing substantial improvements over non-LLM, table-based LLM, and spatio-temporal LLM baselines. Ablation studies and efficiency experiments support the value of each component.

**Questions:**

1. It would strengthen the paper to include qualitative examples of LLM/SLM outputs on different TDP tasks.

2. Clarify whether training on seen tasks uses multitask objectives, and whether different task formats are handled with unified prompt templates or adapters.

3. Discuss robustness to noisy or adversarial input, especially since privacy and integrity are core themes.

4. Specify training data volume and compute cost for the LLM and SLM to better assess scalability.

**Ethical Concerns:**

["NO or VERY MINOR ethics concerns only"]

**Limitations:**

1. While TPA and secret-sharing are presented as privacy-preserving, no formal privacy guarantee (e.g., differential privacy bounds) is provided in the main paper.

2. The actual mechanism of how trajectory knowledge is transferred into the LLM (e.g., dataset format, prompt templates, finetuning details) is high-level. No clear justification for the selected base models (LLaMA, GPT3-Small) beyond empirical performance.

3. Although the framework is positioned as “unified,” it does not explore general trajectory analytics like clustering, forecasting, or anomaly explanation. TDP is a narrow subset of trajectory learning, which limits impact.

4. No analysis or examples are given to explain what the LLM/SLM actually learns, or how/why it succeeds on unseen datasets. This matters given the opaque nature of LLM-based solutions in safety-sensitive applications (e.g., transport systems).

**Quality:**

3

**Strengths And Weaknesses:**

1. Proposes Federated Trajectory Data Preparation (F-TDP), a timely and relevant problem that merges federated learning, trajectory modeling, and LLM adaptation.

2. Well-defined components (TPA, TKE, FPO) address core challenges: privacy, TDP adaptation, and efficiency.

3. Comprehensive evaluation across diverse datasets and tasks, including both seen and unseen settings. Outperforms 13 strong baselines with significant margins.

4. Ablation studies isolate the impact of each module. Communication/runtime benchmarks support claims of scalability and efficiency.

5. The paper acknowledges the privacy/utility tradeoff introduced by TPA and justifies it well.

---

> ### Author Rebuttal · Authors · 2025-07-31
>
> We express our gratitude to the reviewer for providing constructive feedback on our paper, and we greatly appreciate the acknowledgment of our contributions. **We have addressed the specific concerns as detailed below**.
>
> ***Response to Q1&L2&L4:*** Below, we present two LLM/SLM examples for TDP tasks.
> #### **Example 1: Local Trajectory Simplification Task**
> For a local trajectory simplification task, the client's SLM processes a sub-trajectory. The raw data is first normalized using MinMax scaling:
> + Raw Data: [(1201963833, 116.69167, 39.85174), (1201964432, 116.69167, 39.85175), (1201965032, 116.69167, 39.85176), (1201965632, 116.69172, 39.85208), (1201965632, 116.69172, 39.85208), (1201966232, 116.69172, 39.85199), (1201966832, 116.69171, 39.85196), (1201967432, 116.69171, 39.85182)]
> + Normalized Data: [(0.2516, 0.772114, 0.576668), (0.2561, 0.772114, 0.576674), (0.2606, 0.772114, 0.576679), (0.2651, 0.772140, 0.576860), (0.2651, 0.772140, 0.576860), (0.2696, 0.772140, 0.576809), (0.2741, 0.772135, 0.576792), (0.2786, 0.772135, 0.576713)]
>
> The TKE constructs a structured prompt:
> + Task: it is a trajectory simplification task. It aims to reduce the number of spatio-temporal points in a trajectory while preserving its essential shape and features. Data: the trajectory data consisting of spatio-temporal points is [(0.2516, 0.772114, 0.576668), (0.2561, 0.772114, 0.576674), (0.2606, 0.772114, 0.576679), (0.2651, 0.772140, 0.576860), (0.2651, 0.772140, 0.576860), (0.2696, 0.772140, 0.576809), (0.2741, 0.772135, 0.576792), (0.2786, 0.772135, 0.576713)]. Information: none. Format: the output should be the trajectory data.
>
> The SLM processes this prompt and generates:
> + The result of trajectory simplification is [(0.2516, 0.772114, 0.576668), (0.2606, 0.772114, 0.576679), (0.2651, 0.772140, 0.576860), (0.2651, 0.772140, 0.576860), (0.2696, 0.772140, 0.576809), (0.2741, 0.772135, 0.576792)].
>
> It shows that the SLM removed the second and last points.
> #### **Example 2: Cross-Client Trajectory Imputation Task**
> For a cross-client trajectory imputation task, two clients hold sequential sub-trajectories of the same user. The first client has:
> + Raw Data: [(1201955434, 116.49625, 39.9146), (1201956033, 116.50962, 39.91071), (1201956633, 116.52231, 39.91588), ()]
> + Normalized Data: [(0.1886, 0.672904, 0.612130), (0.1931, 0.679691, 0.609936), (0.1976, 0.686134, 0.612852), ()]
>
> The second client has:
> + Raw Data: [(1201957833, 116.59512, 39.90798), (1201958433, 116.61153, 39.88277), (1201959033, 116.65522, 39.8622)]
> + Normalized Data: [(0.2066, 0.723098, 0.608396), (0.2111, 0.731429, 0.594173), (0.2156, 0.753610, 0.582569)]
>
> Each client uses the TPA to encode its sub-trajectory into embeddings sent to the server. For brevity, we denote these embeddings as [p1, p2, p3, ()] for the first client and [p5, p6, p7] for the second client. The server aligns the embeddings and use TKE to construct a prompt:
> + Task: it is a trajectory imputation task. It aims to reconstruct a complete trajectory by predicting or estimating the missing points based on available spatio-temporal data. This often occurs when GPS signals are lost or data collection is interrupted. Data: the trajectory data consisting of spatio-temporal points is [p1, p2, p3, (), p4, p5, p6]. Information: none. Format: the output should be the trajectory data.
>
> The server's LLM processes this prompt and generates:
> + The result of trajectory imputation is [p1, p2, p3, p4, p5, p6, p7].
>
> The result is split and sent back to clients. The first client uses TPA to decode the server's output for p4: (0.2021, 0.707531, 0.612032), with the original format: (1201957233, 116.56446, 39.91442).
>
> ***Response to Q2:*** Multi-task objective and unified prompt template are employed in FedTDP to support generalization across diverse TDP tasks.
> - **Multi-task Objective**. As detailed in **Appendix C.4 (page 20)**, training on the seen tasks in FedTDP is performed using a unified **multi-task learning objective**, where the model jointly learns across all supported TDP tasks (e.g., imputation, map matching, anomaly detection). This allows the model to capture task-shared representations and improve generalization.
> - **Unified Prompt Template**. As introduced in **Section 4.2 (page 5)**, FedTDP uses a standardized prompt format to support multiple TDP tasks. Each prompt includes the fields: **Task, Data, Information, Format**, which together define the task instruction, trajectory input, optional external context (e.g., road network, weather), and expected output format. A variety of examples using this template are provided in **Appendix C.2 (page 18)**.
>
> This unified design **eliminates the need for task-specific adapters or architectures**, and enables flexible and efficient adaptation to new tasks. In the revised version, we will integrate this explanation from the Appendix into the main paper to improve clarity.
>
> ***Response to Q3:*** **Robustness to noisy and adversarial input is a core consideration in the design of our FedTDP**, given its focus on privacy-preserving and reliable trajectory data preparation.
> - **Robustness to Noisy Inputs**. FedTDP is trained and evaluated on real-world datasets (Table 4, page 22), which naturally include noise due to GPS drift, signal interference, and sensor inaccuracies. Importantly, **Noise Filtering is one of the ten core TDP tasks** supported by FedTDP. As shown in **Figs. 5-7**, our method achieves significantly better performance than baselines on the Noise Filtering task, demonstrating its strong capability to detect and correct noisy inputs in practice.
> - **Robustness to Adversarial Inputs**. We provide a **formal theoretical privacy analysis** of the Trajectory Privacy Autoencoder (TPA) in **Appendix C.5 (page 21)**, which uses mutual information to bound potential privacy leakage. This analysis accounts for adversarial scenarios where an attacker may try to infer original data from embeddings, and **proves that the encoded representations are robust against such attacks** when the encoder and decoder remain private.
>
> In summary, FedTDP exhibits strong robustness to both noisy and adversarial inputs through task-specific training (Noise Filtering) and formal privacy guarantees (TPA). To improve clarity and completeness, we will incorporate a discussion on robustness into the main paper in the revised version.
>
> ***Response to Q4:*** Below we provide additional details regarding the training data volume and computational cost to better assess the scalability and practicality of FedTDP.
> - **Training Data Volume**: We use 10% of the GeoLife dataset for training across the 7 seen tasks listed in Table 1. The data is partitioned based on geographical regions and distributed to different clients, simulating a realistic federated setting.
> - **Computational Cost and Efficiency**: As discussed in the efficiency study (**Appendix D.5, page 24**), although LLM-based approaches introduce higher computational costs than non-LLM baselines, FedTDP significantly mitigates this via our Federated Parallel Optimization strategy with **11.3× to 14.2× speedup** compared to other LLM-based baselines.
> - **Computational Complexity**: A formal complexity analysis of the FedTDP framework is provided in **Appendix C.4 (page 21)**, covering both server and client computation.
>
> In summary, FedTDP is explicitly designed to operate efficiently under **resource-constrained environments**, leveraging lightweight SLMs and parallel optimization to ensure **low computation overhead** and **scalable deployment**. In the revised manuscript, we will integrate these critical implementation and scalability details into the main body .
>
> ***Response to L1:*** Actually, **we indeed have provided formal privacy guarantee for TPA in Appendix C.5 (page 21)**. Specifically, we conduct a theoretical privacy analysis using *mutual information* to quantify the upper bound of potential privacy leakage, ensuring a rigorous and quantifiable privacy guarantee when the encoder and decoder remain private.
>
> To avoid misunderstanding, we will move its key theoretical results into the main body of the revised manuscript to improve clarity and emphasize the formal privacy grounding of TPA.
>
> ***Response to L2:*** Actually, **we indeed have provided justification for choosing LLaMA and GPT3-Small thorough empirical study (Appendix D.4, page 28)**. As shown, we evaluated several state-of-the-art LLMs (LLaMA-8B, GPT3-7B, Qwen-7B) and SLMs (GPT3-Small-125M, GPT2-Small-137M, T5-Small-60M) under the FedTDP framework. The results show that:
> - **LLaMA-8B consistently outperforms** GPT3-7B and Qwen-7B in terms of accuracy and generalization.
> - **GPT3-Small-125M achieves the best performance** compared to GPT2-Small and T5-Small.
>
> Therefore, we adopt these models as the default server and client models throughout our experiments. We will integrate a summary of this empirical model selection into the main body to avoid misunderstanding.
>
> ***Response to L3:*** We clarify that the “unified” nature of **FedTDP** is **specifically scoped within the domain of Trajectory Data Preparation**. As also discussed in Section 6, broader downstream analytics tasks such as clustering or forecasting are beyond the current scope of our framework.
>
> TDP focuses on preprocessing tasks including **imputation, map matching, and anomaly detection**, which are crucial for enhancing the trajectory data quality and usability. These tasks are a **prerequisite** for enabling accurate and robust downstream analytics. Without effective TDP, tasks like forecasting or clustering may suffer from corrupted inputs and reduced performance.
>
> Moreover, the selected TDP tasks are **representative, which are widely recognized** in trajectory data communities [1--2] as key challenges.
>
> *[1] Deep Learning for Trajectory Data Management and Mining: A Survey and Beyond, 2024.*
>
> *[2] A Survey on Trajectory Data Management, Analytics, and Learning, 2022.*

---

> > ### Comment · Reviewer_PDos · 2025-08-05
> >
> > Thank you for the detailed and thoughtful rebuttal. I appreciate the additional clarifications and examples provided.
> >
> > That said, I remain unconvinced about the necessity and advantage of using LLMs and SLMs for trajectory data preparation tasks. Many of the tasks described—such as trajectory simplification, imputation, noise filtering, and map matching—are well-established in the literature and have been effectively addressed using traditional methods (e.g., heuristics, probabilistic models, deep neural networks) without the use of generative models.

---

> > > ### Author Response · Authors · 2025-08-05
> > > **Further Clarification for the Confusions Raised by Reviewer PDos**
> > >
> > > ```
> > > That said, I remain unconvinced about the necessity and advantage of using LLMs and SLMs for trajectory data preparation tasks. Many of the tasks described—such as trajectory simplification, imputation, noise filtering, and map matching—are well-established in the literature and have been effectively addressed using traditional methods (e.g., heuristics, probabilistic models, deep neural networks) without the use of generative models.
> > > ```
> > > **We sincerely appreciate the reviewer’s thoughtful feedback and the opportunity to further clarify our methodology**. Below, we address the concerns regarding the necessity of LLMs/SLMs for **trajectory data preparation (TDP)** from three aspects.
> > >
> > > ## 1. LLM Methods vs. Traditional Methods
> > >
> > > **• The Rise of General-Purpose Data Analysis Frameworks**:
> > >
> > > Traditional TDP methods—including probabilistic models, graph neural networks, and heuristic algorithms—have laid a strong foundation for trajectory analysis. However, the rapid evolution of LLMs have demonstrated the superiority of general-purpose frameworks over task-specific solutions. For example, In NLP, **GPT-style LLMs**  [1] now handle *text generation, summarization, translation, and QA* within a single framework, replacing task-specific architectures (e.g., LSTMs for translation, rule-based systems for summarization). In CV, **Vision Transformers (ViTs)** [2] unify *object detection, segmentation, and classification* under one model, surpassing traditional pipelines (e.g., Faster R-CNN for detection, U-Net for segmentation).
> > >
> > > This paradigm shift motivates our proposal for a **unified LLM-based framework for TDP**, capable of handling diverse tasks—including *trajectory anomaly detection, trajectory imputation, noise filtering, stay point detection, map matching, trajectory-user link, trajectory mode identification, trajectory simplification, trajectory segmentation, and trajectory recovery*—within a single architecture.
> > >
> > > **• Emerging LLM-Based Solutions in Trajectory Data Processing**:
> > >
> > > **Recent research has increasingly adopted LLMs to replace conventional trajectory analysis methods**, capitalizing on their superior capacity for learning complex spatio-temporal dependencies. For example, Yang et al. [3] introduced Trajectory-LLM for generating realistic, language-described driving trajectories to enhance motion prediction in autonomous driving scenarios, while Zhou et al. [4] developed TrajCogn to address the challenge of inferring human mobility intentions and travel purposes by interpreting trajectories as narrative sequences. Importantly, as highlighted in our Introduction (Section 1), trajectory data inherently contain sensitive personal information and exhibit cross-domain characteristics, **necessitating robust privacy-preserving analytical approaches**.
> > >
> > > Overall, our work not only aligns with this emerging paradigm but also advances it by specifically tackling the distinctive challenges posed by federated learning environments.
> > >
> > > ## 2. **The Case for LLMs in Trajectory Data Preparation**
> > >
> > > The primary advantage of LLMs lies in their generalization capability across disparate tasks, which addresses critical limitations of traditional approaches:
> > >
> > > •  **Disconnected Pipelines**:
> > >
> > > Current TDP systems rely on specialized models per task. For example, a model trained for map matching cannot assist with anomaly detection, and a noise filtering pipeline must be separately designed and maintained. This leads to deploying various TDP tasks requires training, tuning, and maintaining different separate models, which is **inefficient** and **difficult to scale in real-world systems**.
> > >
> > > •  **Adaptability via Trajectory-Specific Knowledge**:
> > > **To bridge the gap between generic LLMs and trajectory-specific needs**, we introduce the **Trajectory Knowledge Enhancer (TKE)**, which encodes spatio-temporal patterns into the LLM. **Our ablation study (Fig. 8)** shows that removing TKE degrades performance by 27.52%, underscoring its necessity for task adaptation.
> > >
> > > •  **Performance Comparison between our Methods and Traditional Task-Specific Methods**:
> > > Through comprehensive benchmarking against **state-of-the-art specialized methods for each individual trajectory processing task** - including *anomaly detection, imputation, noise filtering, stay point detection, map matching, trajectory-user linking, mode identification, simplification, segmentation, and trajectory recovery* - **our unified framework demonstrates consistent and statistical performance improvements**, as evaluated in Fig. 5 (page 8) with an improvement of at least 18.38%.

---

> > > > ### Author Response · Authors · 2025-08-05
> > > > **Further Clarification for the Confusions Raised by Reviewer PDos**
> > > >
> > > > ## 3. Why SLMs? Efficiency in Federated Deployment
> > > >
> > > > While LLMs offer powerful multi-task learning, **their resource demands necessitate a complementary solution for federated settings**:
> > > >
> > > > •  **Client Constraints**: Deploying LLMs on edge devices is impractical due to computational limits. The SLM, a lightweight variant of the LLM, enables local TDP processing without server dependency.
> > > >
> > > > •  **Collaborative Optimization**: The SLM and LLM are collaboratively optimized via Bidirectional Knowledge Distillation and Trajectory Offsite-Tuning, ensuring that the client model retains strong performance while maintaining efficiency.
> > > >
> > > > ##  Summary
> > > >
> > > > FedTDP combines the **LLM’s generalization** (for server multi-task learning) with the **SLM’s efficiency** (for client execution), creating a unified and privacy-preserving framework. **Again, we deeply appreciate the reviewer’s insightful comments, which have helped us clarify these points**.
> > > >
> > > > We hope the reviewer will find our responses satisfactory and the proposed framework's contributions worthy of consideration for score improvement, given its:
> > > >
> > > > - Novel and Nontrivial integration of LLM/SLM paradigms in trajectory data preparation processing
> > > > - Comprehensive empirical validation across 10 tasks
> > > > - Practical solutions for privacy-preserving federated learning
> > > >
> > > > *[1] Text Alignment Is An Efficient Unified Model for Massive NLP Tasks. NeurIPS 2023.*
> > > >
> > > > *[2] VisionLLM: Large Language Model is also an Open-Ended Decoder for Vision-Centric Tasks. NeurIPS 2023.*
> > > >
> > > > *[3] Trajectory-LLM: A Language-based Data Generator for Trajectory Prediction in Autonomous Driving. ICLR 2025.*
> > > >
> > > > *[4] TrajCogn: Leveraging LLMs for Cognizing Movement Patterns and Travel Purposes from Trajectories. IJCAI 2025.*
> > > >
> > > > All suggestions will be incorporated into the final manuscript. If further clarification would be helpful, we’d be glad to provide it.

---

> > > ### Author Response · Authors · 2025-08-07
> > > **Follow-up on Clarification Regarding LLMs/SLMs**
> > >
> > > Dear Reviewer PDos,
> > >
> > > **Thank you again for your thoughtful comments and for raising the important question** *"regarding the necessity and advantage of using LLMs and SLMs for trajectory data preparation tasks"*.
> > >
> > > **We followed up with a detailed response 2 days ago to further clarify this point and sincerely hope it addresses your concerns**. We understand this is a busy period, but we wanted to gently follow up in case our response was missed.
> > >
> > > If there are any remaining concerns, we would be happy to provide further clarification. **If possible, we would be sincerely grateful if our updated response could be taken into consideration when finalizing the final score**.
> > >
> > > We truly appreciate your time and contributions to improving our work.
> > >
> > > Warm regards,
> > >
> > > The authors of Paper 10076

---

### Official Review · Reviewer_eSCJ · 2025-07-02

**Clarity:** 3
**Significance:** 2
**Originality:** 1
**Rating:** 3
**Confidence:** 4

**Summary:**

This paper proposes a data preparation approach for trajectory data, aiming to protect privacy and the generalizability to different tasks. Specifically, the paper leverages a federated learning framework to train an auto-encoder that converts each trajectory to a low-dimensional representation. Secret sharing is used to protect privacy during model aggregation. The paper further trains large-language models (LLMs) to improve the generalizability of the proposed approach to different tasks. The privacy analysis is done theoretically, and extensive experiments are performed to justify the proposed approach.

**Questions:**

1.	Line 125, the notation D = {D_1, D_2, …} -> {T_1, T_2, …} is not explained. Moreover, it would be more reader-friendly to define what each element exactly means, and how many elements are contained in each set. For example, does T1 represent the first trajectory, and D1 represent the dataset of the first client?

2.	Lines 176-179, the authors mentioned that the advantage of the proposed algorithm over differential privacy (DP) is that DP introduces noise. However, the proposed autoencoder-based strategy essentially also added noise, as the exact trajectory may not be reconstructed from the output of the autoencoder. This undermines the contribution of the proposed method.

3.	The paper seems to have an underlying assumption that the property of a full trajectory (i.e., with its origin and destination in a region) is the same as a partial trajectory (i.e., only part of a full trajectory), so the property of both can be extracted from a single auto-encoder in Section 4.1. However, this may not be true, as travelers make route choice decisions more based on their origin and destination, and the route choice on the full trajectory may not be decomposed into each sub-trajectory.

4.	It is not clear in Section 4.1 whether and how the sub-trajectories of an entire trajectory are aligned across multiple clients.

5.	Minor: Table 1 is a bit hard to read. It is not clear which row in one column corresponds to which row in another column. Please consider reorganizing the table.

**Ethical Concerns:**

["NO or VERY MINOR ethics concerns only"]

**Final Justification:**

This is a good application paper if it is submitted to the application area of transportation, but the methodological contribution is not strong. Hence, my rating is still borderline reject.

**Limitations:**

Yes

**Quality:**

2

**Strengths And Weaknesses:**

Strength

1.	The paper combines multiple techniques, e.g., federated learning, large language models, etc., to address an important engineering problem, i.e., how to prepare trajectory data while preserving privacy.

2.	The proposed method has been extensively tested in a number of existing trajectory datasets and compared to several key benchmarks. The experiment appears solid.

Weakness

1.	The problem setting where trajectory data is divided into different regions is not adequately motivated. The paper mentions “due to legal constraints” , but it is not clear what constraints and why. Also, it is not clear how prevalent the problem setting is. From my understanding, trajectory datasets are more horizontally divided (i.e., each data owner or company has trajectories from a set of individuals) rather than vertically divided (i.e., each dataset contains spatially segmented trajectories).

2.	The designed F-TDP approach seems to be a universal approach that applies not only to trajectory data. In other words, it is not clear what specific properties of trajectory data have enhanced the proposed approach on the privacy part. If there is, please clearly specify.

3.	The methodology of the paper combines multiple techniques. Although the methodology is overall valid, there does not seem to be a strong contribution in each of the technologies or in the synthetic integration of the techniques. Hence, the methodological contribution appears to be relatively weak.

---

> ### Author Rebuttal · Authors · 2025-07-31
>
> Thanks for all the valuable comments and suggestions.
>
> ***Response to W1:*** We provide further clarification regarding **the legal constraints and the motivations behind spatial (vertical) partitioning** of trajectory data in our problem setting:
>
> (1) **Legal Constraints on Centralized Data Aggregation**
>
> Many legal constraints [1--3] mandate data collectors to **minimize non-essential data transmission** and **avoid centralized data storage**, which impose strict limitations on cross-regional data sharing and centralized user tracking, especially when location information can be used to infer identities or behavioral patterns.
>
> (2) **Motivation for Spatial Partitioning**
>
> The spatial/vertical partitioning is highly relevant and realistic in both real-world scenarios and prior research:
> - Real-World Evidence
>   - Uber Movement Platform [4]: It stores trip trajectories locally within their originating administrative regions. Cross-region trips are automatically segmented at regional boundaries, and each segment is stored in its respective region for compliance reasons.
>   - Mobility Data Specification (MDS) [5]: The MDS proposed by Los Angeles Department of Transportation enforces jurisdiction-based data isolation, where different governmental units define boundaries and can only access the data relevant to their region.
> - Prior Research
>   - FedCTQ [6]: It segments user trajectories by geographic region to support privacy-preserving contact tracing in a federated setting;
>   - ATPFL [7]: It uses spatial grids to partition trajectory data and trains region-specific local models, which are then aggregated using federated learning.
>
> *[1] https://www.beijing.gov.cn/zhengce/zhengcefagui/202312/t20231211_3496032.html*
>
> *[2] https://www.gjbmj.gov.cn/n1/2024/0910/c459362-40317200.html*
>
> *[3] https://gdpr-info.eu*
>
> *[4] https://cionews.co.in/data-governance-framework-set-up-by-uber*
>
> *[5] https://github.com/openmobilityfoundation/mobility-data-specification/tree/main/jurisdiction*
>
> *[6] FedCTQ: A Federated-Based Framework for Accurate and Efficient Contact Tracing Query. ICDE, 2024.*
>
> *[7] ATPFL: Automatic Trajectory Prediction Model Design Under Federated Learning Framework. CVPR, 2022.*
>
> ***Response to W2:*** We clarify that the **Trajectory Privacy AutoEncoder (TPA)** is specifically designed to exploit and preserve the **spatio-temporal properties** under privacy constraints.
>
> (1) **Trajectory-Specific Characteristics**
>
> Trajectory data differs fundamentally from generic tabular or image data in the following ways:
> - Temporal regularity: Many trajectories exhibit recurring time patterns (e.g., daily commutes).
> - Spatial dependency: Locations are connected via road networks and geographic proximity.
> - Sequential semantics: The order and timing of visits matter, capturing behavioral patterns and intent.
>
> As a result, standard privacy-preserving methods like differential privacy often disrupt these dependencies.
>
> (2) **How TPA Leverages These Properties**
>
> TPA is specifically designed to preserve these spatio-temporal structures:
> - It encodes each point individually, producing embeddings that retain temporal sequence and spatial locality.
> - It maintains intra-client patterns and inter-client correlations, which are essential for accurate TDP task modeling.
> - Raw trajectory data is never exposed to the global model, ensuring privacy is preserved.
>
> As a result, LLMs in FedTDP can still reason over trajectory patterns without direct access to raw data — a key distinction from generic privacy approaches.
>
> **Overall**, the trajectory-aware design of TPA allows it to both protect privacy and preserve the essential spatio-temporal semantics needed for TDP tasks. This trajectory-specific tailoring is central to the effectiveness of our privacy module.
>
> ***Response to W3:*** We argue that **FedTDP is not a simple combination of existing methods, but a novel, purpose-built framework** for a new and realistic problem setting, **Federated Trajectory Data Preparation (F-TDP)**, which requires **privacy-preserving, efficient, and generalizable TDP across federated clients**. Existing methods fail to meet these demands due to three key unaddressed challenges:
>
> (1) **Preserving Privacy for Cross-Client TDP Tasks**
>
> Traditional FL approaches cannot support contextual reasoning across clients without exposing raw data. Standard methods like differential privacy degrade utility by disrupting spatio-temporal dependencies.
> - In contrast, we propose the **Trajectory Privacy Autoencoder (TPA)**:
>   - It transforms raw trajectories into privacy-preserving spatio-temporal embeddings.
>   - These embeddings retain essential TDP semantics while preventing raw data exposure.
>   - **Theoretical analysis (Appendix C.5, page 21) proves its privacy guarantee** when its encoder/decoder remain private.
>
> (2) **Adapting LLMs to TDP Tasks**
>
> Off-the-shelf LLMs are trained on text and lack understanding of trajectory semantics. Even spatio-temporal LLMs are typically designed for downstream tasks and lack TDP knowledge. To address this, we design the **Trajectory Knowledge Enhancer (TKE)**, which includes:
> - Trajectory Prompt Engineering tailored for TDP semantics.
> - Trajectory Offsite-Tuning and LoRA Sparse-Tuning, which enhances adaptation via lightweight tuning.
> - Bidirectional Knowledge Learning to align SLM/LLM for TDP transfer.
>
> (3) **Improving Efficiency in Resource-Constrained Federated Settings**
>
> LLM deployment in FL is constrained: i) clients lack resources to host LLMs, ii) transmitting all data to the server creates a bottleneck, and iii) even with PEFT, LLM training remains expensive. We address this via the **Federated Parallel Optimization (FPO)**:
> - It deploys a lightweight SLM on the client side and freezes data locally.
> - Enables client-server parallel training, drastically reducing transmission and server load.
> - **Results in 11.3–14.2× runtime speedups (Appendix D.5, 24)**.
>
> **Overall**, each component is carefully designed to address a concrete challenge in F-TDP with theoretical guarantees and empirical validation. This is not a simple stacking of prior methods, but a cohesive and original framework with:
> - A new F-TDP problem definition, grounded in real-world constraints;
> - Three novel modules, each tailored for privacy, adaptation, and efficiency;
> - Extensive experiments across 13 baselines, demonstrating SOTA performance.
>
> We hope this clarifies the novelty and significance of our contributions.
>
> ***Response to Q1:*** D={D_1,D_2,...}->{T_1,T_2,...} denotes all clients' local datasets, where each D_i denotes the set of sub-trajectories held by client i. T_1 denotes the first trajectory in a given client's dataset, and D_1 denotes the entire dataset of the first client. We will incorporate the notation table (**Appendix B, page 17**) into the main text in the revised manuscript for greater clarity.
>
> ***Response to Q2:*** We argue that our **TPA is fundamentally different from DP**, not just in *whether noise is added*, but in the **underlying privacy paradigm**.
>
> (1) **Random Perturbation vs. Learned Transformation**
> - **DP** achieves privacy by adding *random noise* to data, but this **inevitably distorts the data**, breaking critical spatio-temporal correlations (e.g., speed, direction) that are essential for TDP tasks.
> - **TPA**, in contrast, is a **deterministic, learning-based transformation**. It is *not* designed to introduce random noise. Instead, it **filters out privacy-sensitive details** while **preserving useful patterns**.
>
> (2) **DP Limitations in Trajectory-Level Tasks**
>
> While DP is suitable for statistical release or data synthesis, its application to individual trajectory records has **known limitations**:
> - When DP-perturbed trajectories are shared individually (not as aggregates), it is still possible to **infer the original location/time range** probabilistically, within a bound defined by the privacy budget [8,9].
> - Moreover, DP often requires **fine-tuning noise scales**, which leads to a **difficult utility-privacy tradeoff** — particularly problematic in TDP where spatial continuity and temporal granularity are crucial.
>
> (3) **Privacy Guarantee of TPA**
>
> TPA offers a different but **strong privacy guarantee**:
> - The encoder transforms raw trajectories into **privacy-preserving embeddings**.
> - As long as the **encoder and decoder are kept private**, it is **computationally infeasible** to reconstruct the raw trajectory.
> - The mapping is **highly non-linear** and learned from data, and thus resistant to inversion even if the server knows the architecture.
> - This is formally supported by our **theoretical analysis in Appendix C.5 (page 21)**, showing that reconstruction of embeddings is feasible under threat models.
>
> Overall, while both DP and TPA aim to preserve privacy, they operate under **distinct assumptions and mechanisms**.
>
> *[8] Trajectory Privacy Protection Method Based on Differential Privacy in Crowdsensing. TSC, 2024.*
>
> *[9] Privacy Preservation for Trajectory Publication Based on Differential Privacy. TIST, 2022.*
>
> ***Response to Q3:*** Actually, **we do not assume that full and partial trajectories share the same properties and sub-trajectories are independent**. Actually, in our problem setting, a full trajectory may span multiple regions, as shown in Fig. 1 (page 2). Due to privacy constraints, when a trajectory crosses regional boundaries, it is naturally segmented to sub-trajectories where each is stored locally at its respective region.
> + Local TDP tasks are performed independently on the sub-trajectory, relying solely on its spatio-temporal patterns.
> + Cross-region TDP tasks require sub-trajectories to be reconstructed into a trajectory with the full movement context for processing.
>
> ***Response to Q4:*** Sub-trajectories from the same user across clients are aligned via the anonymized user identifier.
>
> ***Response to Q5:*** We will reorganize Table 1 to clearly align rows and enhance clarity.

---

> > ### Comment · Reviewer_eSCJ · 2025-08-05
> >
> > Thank you for the clarifications.
> >
> > W1. The authors have provided several examples for the problem settings where trajectory data is partitioned spatially. However, reference [4] does not work, and I do not find the relevant information in reference [5].
> >
> > W3. I agree that the paper is good in terms of application, but the contributions, as mentioned by the authors, are not strong in the methodological sense. Therefore, I will keep my original rating.

---

> > > ### Author Response · Authors · 2025-08-06
> > > **Further Clarification for the Confusions Raised by Reviewer eSCJ**
> > >
> > > Thank you for the follow-up feedback. We sincerely appreciate your careful reading of our work and your willingness to engage in this discussion. **We believe that we have addressed all the follow-up concerns**.
> > >
> > >
> > > ```
> > > W1. The authors have provided several examples for the problem settings where trajectory data is partitioned spatially. However, reference [4] does not work, and I do not find the relevant information in reference [5].
> > > ```
> > >
> > > We sincerely thank you for your attention to the links of real-world scenarios. **Below, we quote directly from the sources to clarify how they support our spatially partitioned (vertical) trajectory data setting**.
> > >
> > > ### **(1) Uber Movement Platform [4]**:
> > >
> > > The article [4] describes Uber’s data governance strategy, which explicitly acknowledges regional differences in data policies and the need for localized data handling:
> > >
> > > - As quoted, "Even though the data is collected globally, the governance that it needs to apply is different based on regions from where the data is coming from." This directly reflects the reality that trajectory data generated across different regions (e.g., cities or countries) is subject to distinct data policies and cannot be freely shared.
> > > - The statement *"We make changes in the systems as per the changing governance and policies so that customers and partners on our platform are compliant to the policies." further emphasizes the need for system designs that respect regional data boundaries, which is precisely the motivation for our federated, privacy-preserving framework.*
> > >
> > > ### **(2) Open Mobility Foundation - Jurisdiction API [5]**:
> > >
> > > The Open Mobility Foundation’s Mobility Data Specification (MDS) [5] design the Jurisdiction API specification to allow cities to define geographic boundaries and enforce data policies (e.g., data access, retention, sharing) within their jurisdiction：
> > >
> > > - The quote *"For a single jurisdiction MDS deployment, a city designates a jurisdiction that providers can reference and know in what area to send events. When a trip leaves the LADOT jurisdiction, providers need to send an event with the vehicle state set to elsewhere."* illustrates that a single trajectory is naturally segmented at jurisdictional borders.
> > > - Furthermore, *"City and transportation agencies need to regulate mobility within their own jurisdictions. Within a collection of agencies under a single MDS software deployment, those agencies need to coordinate and share relevant data between one another when their jurisdictions overlap."* highlights both the necessity of cross-region collaboration.
> > >
> > > These evidences imply that mobility data, including trajectories, must be processed and governed according to the specific region (jurisdiction) in which it was generated. Cross-region data sharing requires explicit policy alignment, which is often restricted. This supports our assumption that clients (regions) hold local trajectory data and cannot share raw data freely.
> > >
> > > *[4] https://cionews.co.in/data-governance-framework-set-up-by-uber*
> > >
> > > *[5] https://github.com/openmobilityfoundation/mobility-data-specification/tree/main/jurisdiction*

---

> > > ### Author Response · Authors · 2025-08-06
> > > **Further Clarification for the Confusions Raised by Reviewer eSCJ**
> > >
> > > Thank you for the follow-up feedback. We sincerely appreciate your careful reading of our work and your willingness to engage in this discussion. **We believe that we have addressed all the follow-up concerns**.
> > >
> > > ```
> > > W3. I agree that the paper is good in terms of application, but the contributions, as mentioned by the authors, are not strong in the methodological sense. Therefore, I will keep my original rating.
> > > ```
> > >
> > > We sincerely appreciate your recognition of the application value of our work. While we respect your assessment, we would like to clarify that we do see strong methodological contributions in our work — a perspective that also appears to be acknowledged by the other reviewers, who recognized the technical novelty of our approach.
> > >
> > > In particular, we firmly believe that our framework presents substantial methodological innovations, not only in each individual component but also in their co-design to address the new and practical problem of Federated Trajectory Data Preparation (F-TDP). These innovations tackle interdependent challenges that, to the best of our knowledge, have not been jointly addressed in prior work, and are supported by both theoretical analysis and experiments.
> > >
> > > Below, we summarize the core methodological contributions where each component is tailored to the unique demands of F-TDP, and supported by theoretical analysis and extensive experiments.
> > >
> > > ### **(1) Novel Problem Formulation: F-TDP**
> > >
> > > We formally define F-TDP, **a new problem where trajectory data is spatially partitioned across clients**, and TDP tasks must be performed without raw data sharing. This setting is grounded in real-world constraints (as shown above), **which requires cross-client trajectory reasoning under privacy, efficiency, and generalization constraints**.
> > >
> > > ### **(2) Three Novel, Interdependent Modules**
> > >
> > > **• Trajectory Privacy Autoencoder (TPA):**
> > >
> > > - Not just an autoencoder: TPA is designed to preserve spatio-temporal semantics (e.g., continuity, speed, direction) in embeddings while preventing raw trajectory reconstruction.
> > > - **Theoretical privacy guarantee**: We prove in Appendix C.5 (page 21) that when the encoder/decoder are private in clients, the mutual information $I(T;E)$ between raw trajectory $T$ and embedding $E$ is bounded, making inversion attacks infeasible.
> > > - Superior to DP that adds noise and degrades utility, TPA maintains high data utility, which is critical for TDP tasks.
> > >
> > > **• Trajectory Knowledge Enhancer (TKE):**
> > >
> > > - Not a plug-and-play LLM: Generic LLMs fail on TDP tasks (see Fig. 6–7, page 8). TKE introduces:
> > >   - Trajectory prompt engineering to encode spatio-temporal semantics.
> > >   - Trajectory offsite-tuning and sparse-tuning for efficient and federated adaptation.
> > >   - Bidirectional knowledge learning to align SLM and LLM for TDP-specific knowledge transfer.
> > > - **This enables multi-task generalization across 10 diverse TDP tasks, which is a capability absent in traditional or even spatio-temporal LLMs**.
> > >
> > > **• Federated Parallel Optimization (FPO):**
> > >
> > > - A lightweight SLM is deployed on each client for local TDP tasks, reducing server workload and leveraging clients’ computational resources.
> > > - Frozen data decouple client and server training, enabling asynchronous and parallel optimization.
> > > - **This design drastically reduces communication and computation overhead, achieving 11.3–14.2× faster training than LLM-based baselines (Appendix D.5, page 24)**.
> > >
> > > ### **(3) Experimental Validation of Novelty and Impact**
> > >
> > > Our framework is not just conceptually novel—**it outperforms 13 state-of-the-art baselines across 6 real-world datasets and 10 TDP tasks**, with improvements ranging from 4.84% to 45.22% (Figs. 5-7, page 8).
> > >
> > > Ablation studies (Fig. 8, page 9) confirm the necessity of each component:
> > >
> > > - Removing TPA → Performance remains relatively stable.
> > > - Removing TKE → 27.52% performance drop.
> > > - Removing FPO → 4 times slower training.
> > >
> > > These results empirically validate that our integration is not arbitrary, but essential for solving F-TDP.
> > >
> > > ### **Response Conclusion**
> > >
> > > FedTDP’s co-design — **TPA for privacy-aware context sharing**, **TKE for task-generalized LLM adaptation**, and **FPO for efficient parallel training** — represents a significant methodological advance in federated learning for trajectory data preparation. We believe this framework offers both **practical relevance** and **technical innovation**, tackling a novel and challenging problem space.
> > >
> > > We hope that the above clarifications and the experimental evidence may help convey the strength of our methodological contributions more clearly. **We would be grateful if this could inform a reconsideration of the overall evaluation**. All suggestions will be incorporated into the final manuscript. If further clarification would be helpful, we’d be glad to provide it. Again, we greatly appreciate the reviewer’s thoughtful feedback and the time dedicated to reviewing our work!

---

> > > ### Author Response · Authors · 2025-08-08
> > > **Response to the “Response to Further Comment from Reviewer eSCJ”**
> > >
> > > Dear Reviewer eSCJ,
> > >
> > > Thank you once again for your thoughtful comments and for raising the important concerns below following our rebuttal.
> > >
> > > *W1. The authors have provided several examples of problem settings where trajectory data is partitioned spatially. However, reference [4] does not work, and I cannot find the relevant information in reference [5].*
> > >
> > > *W3. I agree that the paper is strong in terms of application, but as the authors' stated contributions are not particularly strong from a methodological perspective, I will keep my original rating.*
> > >
> > > **We have submitted a detailed response 3 days ago to further clarify these points**, and we sincerely hope it addresses your concerns. We understand this is a particularly busy time, but we wanted to follow up gently in case our response was overlooked.
> > >
> > > **If there are any remaining issues, we would be more than happy to provide additional clarification. If possible, we would be truly grateful if our updated response could be taken into consideration when finalizing the review decision**.
> > >
> > > We greatly appreciate your time and effort in helping us improve our work.
> > >
> > > Warm regards,
> > >
> > > The authors of Paper 10076

---

### Official Review · Reviewer_zmKm · 2025-07-03

**Clarity:** 3
**Significance:** 2
**Originality:** 2
**Rating:** 4
**Confidence:** 3

**Summary:**

- The paper proposes a privacy preserving framework for trajectory data preparation (Trajectory Privacy AutoEncoder), utilizing federated learning, multi-party computation to prevent information leakage from embeddings.
- The paper uses LLM models (Trajectory Knowledge Enhancer) to understand trajectory data, and apply on issues such as noise filtering and stay point detection.
- As a last step, Federated Parallel Optimization improves training efficiency by minimizing data transmission and enabling parallel execution.
- The authors demonstrate that the given setup out-performs state-of-the-art baselines, improving performance ranging from 4.84% to 45.22% across 10 mainstream TDP tasks.

**Questions:**

- How does introducing LLM affect training and inference speed compared to baselines?
- How does federated learning + multi-party computation affect the computation overhead? How many rounds of communication are required and what is the communication data size?
- How are the LLM baselines applied to TDP related tasks, since many of them are generic spatio-temporal understanding tasks?

**Ethical Concerns:**

["NO or VERY MINOR ethics concerns only"]

**Limitations:**

yes

**Quality:**

3

**Strengths And Weaknesses:**

Strength
- The paper unifies trajectory data planning with privacy centric designs and LLM inferences. The framework is also versatile to multiple TDP related tasks.
- Performance improvement over baselines are evident and clearly presented. The clarity of paper writing is excellent and ideas are well-presented.


Weakness
- The paper lacks discussion on the computation overhead introduced by the additional measurements (e.g. LLM, privacy preserving protocols).
- It’s unclear whether the performance improvement is a result of superior framework, or LLM’s inherent capability. The non-LLM baselines can’t leverage strong contextual capabilities of the LLMs, while the LLM baselines are not TDP specific, and it’s unclear how they are applied to TDP tasks.

---

> ### Author Rebuttal · Authors · 2025-07-31
>
> We express our gratitude to the reviewer for providing constructive feedback on our paper, and we greatly appreciate the acknowledgment of our contributions. **We have addressed all the concerns as detailed below**.
> ```
> W1. The paper lacks discussion on the computation overhead introduced by the additional measurements (e.g. LLM, privacy preserving protocols).
> Q1. How does introducing LLM affect training and inference speed compared to baselines?
> Q2. How does federated learning + multi-party computation affect the computation overhead? How many rounds of communication are required and what is the communication data size?
> ```
> Thanks for the insightful comments and questions regarding the computational overhead introduced by the additional measurements (e.g. LLM, privacy preserving protocols). We appreciate the opportunity to clarify.
>
> - **Regarding the overhead introduced by LLMs**, we have indeed conducted a detailed efficiency study, as presented in **Appendix D.5 (page 24)**. While LLM-based methods naturally incur higher training and inference costs due to their large parameter size and autoregressive nature, our proposed FedTDP framework **significantly alleviates this overhead**. Specifically, the **FPO (Federated Prompt Optimization) module** improves system efficiency by reducing redundant data transmission and enhancing training parallelism. Our results show that FedTDP achieves **11.3× to 14.2× faster runtime** compared to standard LLM-based baselines, without compromising accuracy. This demonstrates that integrating LLMs, when combined with FPO, can be efficient and scalable.
>
> - **Regarding the privacy-preserving mechanism overhead**, we carefully designed the **TPA (Trajectory Privacy AutoEncoder)** and its integration with **decentralized secret-sharing** to minimize cost:
> + As described in **Section 4.1 (page 4)**, the TPA module is a lightweight three-layer MLP, contributing negligible computation burden. In addition, the adopted **secret-sharing-based aggregation** (Eqs. 2–3, page 5) avoids costly cryptographic primitives like homomorphic encryption or secure multi-party computation, ensuring low runtime complexity.
>
> + Our **ablation study (Section 5.2, page 9)** further supports this. Removing TPA results in only marginal gains in runtime and communication size, indicating that it is lightweight. Quantitatively, **TPA introduces ~5GB additional communication** over 10 rounds, while the total communication cost of FedTDP is ~40GB — most of which comes from model and embedding exchanges, not the privacy module.
>
> In summary, while both the LLM integration and the privacy-preserving mechanism introduce some additional overhead, our design ensures this overhead is carefully controlled, and the overall system remains efficient.
>
> ```
> W2. It's unclear whether the performance improvement is a result of superior framework, or LLM's inherent capability. The non-LLM baselines can't leverage strong contextual capabilities of the LLMs, while the LLM baselines are not TDP specific, and it's unclear how they are applied to TDP tasks.
> Q3. How are the LLM baselines applied to TDP related tasks, since many of them are generic spatio-temporal understanding tasks?
> ```
> Thank you for the thoughtful question. We clarify that the performance improvements of FedTDP do not solely stem from LLMs' inherent capabilities, but primarily from our proposed architecture, which is specifically tailored to trajectory data processing (TDP) tasks. The core innovation lies in the Trajectory Knowledge Enhancer (TKE) module.
>
> #### **(1) Clarification on LLM vs. Framework Contribution**
>
> As shown in **Figs. 6–7 (Page 8)**, while LLM-based baselines (e.g., directly inputting raw trajectory data into pre-trained LLMs) exhibit slight improvements over non-LLM models, **their overall performance remains limited**, since they lack task-specific design for TDP. In contrast, **FedTDP outperforms all baselines by a significant margin — with gains ranging from 4.84% to 45.22% over LLM baselines**. This gap demonstrates that the improvement is not just due to LLM capability, but the effectiveness of our framework design, especially TKE.
>
> #### **(2) How We Adapt LLMs to TDP Tasks**
>
> To bridge the gap between generic LLMs and TDP-specific requirements, we introduce the **Trajectory Knowledge Enhancer (TKE)**, which enables LLMs to understand and reason over trajectory data via:
>
> - **Trajectory Prompt Engineering**: We craft a structured prompt schema with four fields — Task, Data, Information, and Format — to provide TDP-specific semantics to the LLM.
> - **Trajectory Tuning**: We introduce lightweight adapters (e.g., LoRA-style fine-tuning) for efficient LLM adaptation to TDP.
> - **Bidirectional Knowledge Learning**: We enable mutual knowledge enhancement between LLM and a lightweight SLM to reinforce TDP reasoning capabilities.
> - As demonstrated in **ablation study**, removing TKE causes a dramatic drop in performance (≥ 27.52%), underscoring its critical role.
>
> **In summary**, FedTDP's improvement is not merely due to LLM integration but is primarily driven by a novel, TDP-specific framework that enables effective use of trajectory semantics within large models.

---

> ### Author Response · Authors · 2025-08-06
> **Kind Reminder: Request for Discussion on Our Rebuttal**
>
> Dear Reviewer zmKm,
>
> **We sincerely appreciate your time and insightful comments on our manuscript. Your expertise has been invaluable in helping us improve this work, and we're truly grateful for your positive recognition of our work**.
>
> We sincerely apologize for the urgency of our follow-up as the deadline approaches. Please know we would not reach out unless absolutely necessary, and we truly regret any inconvenience this may cause.
>
> As we deeply respect your professional perspective, we would be most grateful if you could kindly share whether our responses have adequately addressed your concerns. Should any points benefit from further clarification, we would be delighted to provide additional details to ensure all aspects meet your high standards. However, we completely understand if the timeline makes this difficult.
>
> Thank you again for your exceptional guidance throughout this process. Your professional insights have significantly elevated our paper, and we're profoundly grateful for your contribution to our work.
>
> With utmost respect,
>
> The Authors of Paper 10076

---

### Note · Authors · 2025-08-14

We thank the reviewers for their constructive engagement. Our rebuttal comprehensively addressed all concerns with empirical evidence, theoretical analysis, and concrete revision plans. Below is a concise reviewer-by-reviewer summary for the AC:

- **Reviewer zmKm** (score: 4, conf: 3) stated our rebuttal was ***detailed*** and ***addressed their concerns***, keeping ***borderline accept***.

- **Reviewer PDos** (score: 4, conf: 3) described our rebuttal as ***detailed and thoughtful***, maintaining ***borderline accept***.

- **Reviewer eSCJ** (score: 3, conf: 4) raised: (W1) motivation for spatially partitioned trajectory data; (W3) weak methodological contribution.

  - **W1**: We cited real-world platforms (Uber Movement, MDS Jurisdiction API) and regulations (Chinese law, GDPR) showing cross-region trajectories are segmented and stored locally due to jurisdictional policies—confirming our setting is realistic and prevalent.
  - **W3**: We clarified that FedTDP is a purpose-built framework for the **new F-TDP problem**, addressing three interdependent and non-trivial challenges via three carefully designed components:
    - **TPA** — trajectory privacy autoencoder preserving spatio-temporal semantics with theoretical guarantees.
    - **TKE** — LLM extension/modification for TDP semantics and cross-task generalization.
    - **FPO** — federated parallel optimization achieving 11.3–14.2× training speedup.
    - **All components have theoretical grounding and superior experiments across SOTA 13 baselines, 6 datasets, and 10 TDP tasks**.

  In rebuttal, the reviewer also stated ***the paper is good in terms of application***. While their score was unchanged, our clarifications and evidence demonstrate both methodological strength and practical relevance.

- **Reviewer P2Gk** (score: 3, conf: 4) acknowledged that ***“The authors addressed some of my concerns”***. Their issues—**novelty, figure clarity, requested details**—were resolved. Note that, the **novelty** is recognized by other reviewers while the rest **figure clarity and requested details** are presentation/implementation matters we have revised.

**In summary**, our rebuttal offered **comprehensive, evidence-backed clarifications** grounded in real-world applicability, theoretical rigor, and extensive experimentation (**13 baselines, 6 datasets, 10 TDP tasks**). We believe **FedTDP** is a novel, practical, and rigorously validated solution for privacy-preserving federated TDP.

---

### Decision · Program_Chairs · 2025-09-17

**Decision:**

Reject

**Comment:**

This paper proposes a federated learning framework for trajectory data preparation, and four reviewers have submitted their recommendations for the manuscript. The paper performs a unified combinition of federated learning, LLM and trajectory data preparation. However, main concerns for this work are the lack of novelty, unfair baseline comparisons, and lack of privacy analysis. Half of reviewers feel that the quality aren't above average, after reading the authors' feedback. During the discussion period, no one has argued for an acceptance.